# A social niche breadth score reveals niche range strategies of generalists and specialists

F. A. Bastiaan von Meijenfeldt ⬮ [1,3], Paulien Hogeweg[1] & Bas E. Dutilh ⬮ [1,2] ✉

Generalists can survive in many environments, whereas specialists are restricted to a single environment. Although a classical concept in ecology, niche breadth has remained challenging to quantify for microorganisms because it depends on an objective definition of the environment. Here, by defining the environment of a microorganism as the community it resides in, we integrated information from over 22,000 environmental sequencing samples to derive a quantitative measure of the niche, which we call social niche breadth. At the level of genera, we explored niche range strategies throughout the prokaryotic tree of life. We found that social generalists include opportunists that stochastically dominate local communities, whereas social specialists are stable but low in abundance. Social generalists have a more diverse and open pan-genome than social specialists, but we found no global correlation between social niche breadth and genome size. Instead, we observed two distinct evolutionary strategies, whereby specialists have relatively small genomes in habitats with low local diversity, but relatively large genomes in habitats with high local diversity. Together, our analysis shines data-driven light on microbial niche range strategies.

Culture-independent sequencing studies have greatly expanded our understanding of the microbial world. They uprooted the tree of life[1,2], revolutionized our view of the human microbiome and virome[3,4] and advanced our comprehension of early evolution[5,6]. By using standardized protocols across large numbers of samples[7–10], classical ecological questions can now be addressed on the global scale. A quintessential question is that of ecological niche breadth[11,12]—the range of conditions under which an organism can live. Although the distinction between specialists and generalists is a fundamental property of life and its evolution, general mechanisms that determine niche breadth are poorly understood[13] and quantification has proven challenging[14].

Microbial niche breadth has been measured for specific aspects of the environment (for example, temperature[15,16], pH[17] and nutrient dependence[18,19]). Niche breadth definitions that assess the full *n*-dimensional niche space[20] have been based on occurrence in environmental samples. Rather than the theoretical fundamental niche, microbial occurrence represents its empirical realized niche. Because of complex interactions within microbial communities, the realized niche can be both smaller (for example, due to competition[21]) or larger (for example, due to metabolic dependencies[22]). Previous studies defined organisms that are present in many samples or predefined habitats as generalists, and rare organisms as specialists[23–26]. Based on this definition, Sriswasdi et al.[27] suggested an important evolutionary role for generalist species in maintaining taxonomic diversity, with generalists having higher speciation rates and persistence advantages over specialists. Others defined the niche breadth of an organism by the uniformity of its distribution across habitats[28], suggesting that community assembly of specialists is driven by deterministic processes, whereas for generalists neutral processes are more important[29,30]. Notwithstanding these intriguing results, niche breadth studies based on occurrence in microbiomes have been sensitive to biases due to habitat definition and sample selection.

Microbiomes are sensitive biomarkers capable of detecting geochemical gradients[31], host health status[32–34] and metabolites in a given

[1]Theoretical Biology and Bioinformatics, Department of Biology, Science for Life, Utrecht University, Utrecht, the Netherlands. [2]Institute of Biodiversity, Faculty of Biological Sciences, Cluster of Excellence Balance of the Microverse, Friedrich Schiller University Jena, Jena, Germany. [3]Present address: Department of Marine Microbiology and Biogeochemistry, NIOZ Royal Netherlands Institute for Sea Research, Texel, the Netherlands. ✉e-mail: bedutilh@gmail.com

niche[35,36]. We thus reason that the vast collection of tens of thousands of environmental sequencing datasets that are available in the public domain[37] could be used to implement an unbiased, data-driven and comprehensive niche breadth definition, based on community similarity between samples where microbial taxa occur. As such, we treat community composition as a proxy for the realized niche of a microorganism that reflects both the abiotic environment and the microbial interactions within. Similar reasoning has been used to quantify the niche range of eukaryotes without the use of external habitat definitions[38]. In this view, organisms that occur in compositionally similar samples are social specialists, as their niche is restricted to the same local neighbours, and organisms that occur in compositionally dissimilar samples are social generalists, as they are more flexible in their interaction partners. Using community similarity as a substitute for ecological range, we developed a social niche breadth (SNB) score that allowed us to quantify the social niche range for taxa at all taxonomic ranks and assess strategies for specialization and niche range expansion throughout the prokaryotic tree of life.

## Results

### SNB captures global heterogeneity in microbial communities

To compare the niche breadth of microbial taxa, we devised and extensively benchmarked (Supplementary Information) an SNB score that exploits the abundantly available meta-omics datasets derived from diverse environments around the world (Fig. 1 and Supplementary Data 1 and 2). These microbiomes are taxonomically annotated with the same MGnify pipeline[37], which allows for a comparison of vastly different environments, studies and experiment types (Supplementary Information). First, we assessed the biome annotations of these datasets, as provided by the dataset submitters. The annotations highlighted the main drivers of microbiome composition (Fig. 1d and Supplementary Information), including salinity (t-distributed stochastic neighbour embedding dimension 1 (t-SNE 1)) and host association (t-SNE 2)[10,39–41]. The 22,518 samples covered a total of 140 annotated biomes that differed markedly in within-sample (α) and between-sample (β) diversity. Annotated biomes with high mean α diversity, such as soils, had low β diversity (Fig. 1f), implying a relatively stable core community across these high-diversity habitats.

Most samples from the same annotated biome are relatively similar, as reflected by a low β diversity. Nevertheless, annotated biome definitions are arbitrarily delineated and may be subject to human error. For example, the plants biome includes both freshwater plants[42–44] and seagrasses[45], as well as macroalgae from kelp forests[46] (Supplementary Data 1). Also, it is difficult to quantify the degree of similarity between categorical biomes in a biologically meaningful way. We used the observation that microbiomes are biomarkers[31–36] and developed SNB, which captures the compositional heterogeneity of samples for which a taxon is found to quantify niche breadth.

We assume that the small subunit rRNA gene that is queried is a proxy for the genetic content of a taxon that defines its traits. Specific traits exist at all taxonomic ranks and determine their occurrence across microbiomes[47]. Since the taxonomic annotations are based on a reference taxonomy and the biosphere is relatively unexplored, sometimes high-ranking taxa do not have low-ranking annotations like species (Fig. 1e). We considered that members of a taxon are alive and growing if the taxon represented a relative abundance of at least 1/10,000 of the prokaryotic reads in a sample, and thus ignored the possibility of migration from other sources and the potential for dead organic matter contributing DNA to the sequencing results[48]. Next, we defined SNB as the mean pairwise dissimilarity between these microbiomes. After benchmarking 150 different ecological dissimilarity measures for their ability to separate the annotated biomes, we chose mean pairwise dissimilarity based on the inverse Spearman's rank correlation of known taxa at taxonomic rank order to quantify SNB (Supplementary Information). Thus, taxa with a low SNB score are found in

samples with very similar microbial composition (social specialists) and taxa with a high SNB score are found in dissimilar samples (social generalists). Our approach accounts for database biases, as some environments are much more frequently sampled than others (Fig. 1a and Supplementary Information). Indeed, taxa that are detected in the same number of samples or annotated biomes may have very different SNB scores (Fig. 2a–c). Different from studies that investigate the co-occurrence of taxa across samples[49,50], SNB quantifies the range of communities that a taxon can occur in. SNB treats each sample as a local niche and infers that taxa that occur across highly differing communities are social generalists, while taxa that occur in similar communities are social specialists. Since SNB is calculated when a taxon is present over a detection limit of 1/10,000 reads, the relative abundance and associated variability of a taxon's distribution are observables and can be associated with niche range rather than part of the definition, as in ref. 28.

### SNB throughout the prokaryotic tree of life

To investigate the distribution of social generalists and specialists throughout the prokaryotic tree of life, we calculated SNB for taxa at all ranks (Fig. 2d and Supplementary Data 3). For the vast majority of taxa, the SNB score is lower than expected based on random permutations (Fig. 2a–c), indicating that all microorganisms are social specialists to some extent because they occur in a non-random subset of all samples. Exceptions to this rule include the high-ranking superkingdom Bacteria and phylum Proteobacteria, which are widespread, occurring in 22,295 and 22,211 of the 22,518 samples, respectively. While there is a clear positive correlation between SNB and the number of samples in which a taxon occurs (Fig. 2a–c), very rare taxa such as *Aminobacter* (five samples) and *Methanimicrococcus* (28 samples) still have a high SNB (SNB = 0.56 and SNB = 0.51, respectively). Alternatively, some taxa that are found in many samples have a relatively low SNB because these samples are very similar in composition (for example, *Phyllobacterium* (226 samples; SNB = 0.03) and *Geminicoccus* (473 samples; SNB = 0.09)).

The distribution of SNB scores differs per taxonomic rank. High-ranking taxa tend to have higher SNB scores than low-ranking taxa (Fig. 2d,e), which intrinsically occur in a subset of the samples of their parent taxa. High-ranking taxa can have high SNB scores either because they contain subtaxa that are specialists in different communities or because the subtaxa are also generalists. To compare taxa at different ranks, we calculated a rank-specific modified $z$ score (Fig. 2d and Supplementary Data 4), where positive $z$ scores indicate that the SNB of the taxon is higher than the median for its rank and the taxon is thus relatively generalist and negative $z$ scores indicate that it is relatively specialist. For example, the family Flavobacteriaceae and the genus *Prevotella* are social generalists (with $z$ scores of 2.02 and 0.61, respectively), but their subtaxa are relatively specialized for their rank (median $z$ score of genera in Flavobacteriaceae = −0.53; median $z$ score of species in *Prevotella* = −0.71). The family Lactobacillaceae on the other hand is generalist ($z$ score = 0.46) and its genera are also generalists (median $z$ score = 1.73). In addition, high-ranking taxa with high SNB scores often have more subtaxa than high-ranking taxa with low SNB scores (Supplementary Fig. 16). This suggests that the diversity of taxa, as currently represented by taxonomy, reflects their ecological range well. The four best-studied phyla, Proteobacteria, Firmicutes, Bacteroidetes and Actinobacteria, which together cover 97% of cultured prokaryotic species[51], are dominant across a wide range of environments (Supplementary Fig. 17) and have a higher SNB than others (Fig. 2d).

There are many phyla that have low SNB scores and contain few classes, orders and families compared with the dominant ones described above. Subtaxa of these low-scoring phyla are thus under-represented at the class, order and family ranks and we observe that the distribution of SNB scores is more skewed towards social specialism at the phylum rank (median SNB = 0.36) than at these lower ranks (median SNB = 0.38–0.43; see Fig. 2d). Many phyla with

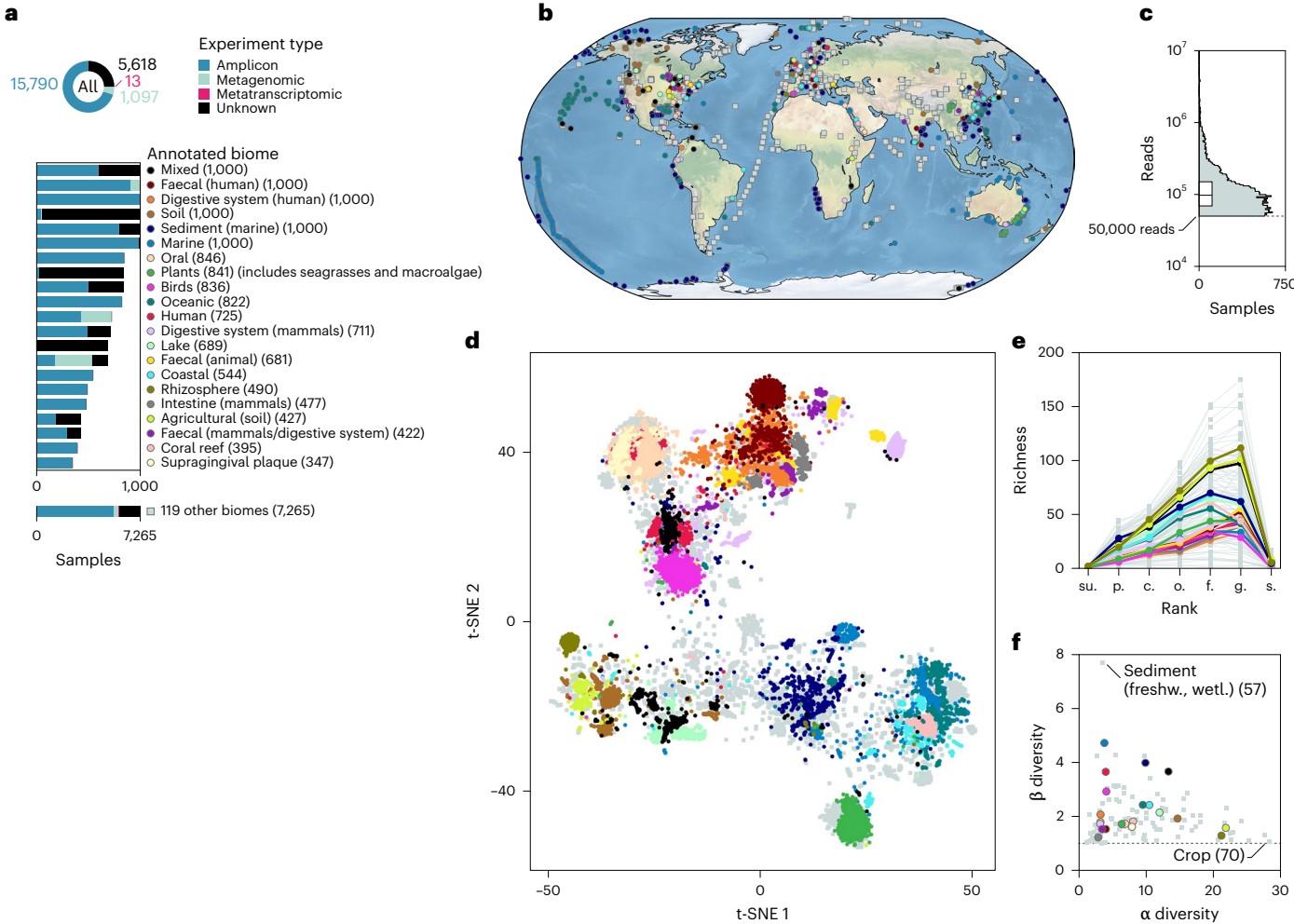

**Fig. 1 | A diverse and global microbial dataset. a**, Samples were received from vastly different annotated biomes and study designs. The numbers in parentheses indicate the number of samples within the annotated biome. Annotated biomes with fewer than 347 samples have been grouped as other. For a hierarchical tree of all annotated biomes, see Supplementary Fig. 1b. **b**, Geographical distribution of the samples. **c**, Total number of taxonomically annotated reads per sample ($n$ = 22,518 samples). The box plot shows the interquartile range and median. No samples with fewer than 50,000 reads were selected. **d**, Samples from similar annotated biomes cluster together based on taxonomic profile in a t-SNE visualization (perplexity = 500), with the same ecological dissimilarity measure used as for SNB (namely, the Spearman's rank correlation coefficient ($0.5 - (\rho/2)$) of known taxa at taxonomic rank order). For a PCoA visualization of the same data and the positions of all 140 annotated biomes on the PCoA, see Supplementary Figs. 2 and 3, respectively. Most samples from the plants biome were derived from seagrasses and macroalgae from kelp forests. **e**, Taxa richness differs per annotated biome and taxonomic rank. The low number of annotated species is a consequence of a relatively unexplored biosphere. su., superkingdom; p., phylum; c., class; o., order; f., family; g., genus; s., species. **f**, Annotated biomes with high mean α diversity have low β diversity, whereas both low and high β diversity is found among annotated biomes with low mean α diversity. freshw., freshwater; wetl., wetlands.

the *Candidatus* status have a low SNB compared with validly described phyla (Supplementary Fig. 17). The connection between the *Candidatus* status and low SNB may reflect a discovery bias of these phyla where widespread lineages tend to be discovered and described sooner than rare ones, although some candidate phyla are widespread (Supplementary Fig. 17). Candidate phyla may require specific growth conditions, which can be reflected in relatively stable specialized microbial communities, consistent with their low SNB. In addition, several candidate phyla, including the bacterial candidate phyla radiation and DPANN archaea, may consist of obligate symbionts of specific hosts[2]. Whereas it was recently shown that consortia of obligate symbionts can grow on a wider range of carbon sources than their individual members and thus expand their metabolic niche[22], the individual microorganisms in these consortia are social specialists as they require specific partners in their local communities.

Taxa with high and low $z$ scores are dispersed throughout the prokaryotic tree of life (Fig. 2d), indicating that social specialization

and niche range expansion happened independently numerous times in evolution. Phyla with relatively specialized genera include Proteobacteria (median $z$ score = −0.07), Bacteroidetes (median $z$ score = −0.26), Actinobacteria (median $z$ score = −0.17), Cyanobacteria (median $z$ score = −0.72), Planctomycetes (median $z$ score = −0.37), Acidobacteria (median $z$ score = −0.47) and Chloroflexi (median $z$ score = −0.17), whereas Firmicutes, Tenericutes and Euryarchaeota have genera that are relatively generalist (median $z$ scores of 0.43, 1.18 and 1.06, respectively). Taxa with relatively low SNB for their ranks include known specialists such as the genus *Christensenella*[27] ($z$ score = −1.01), but also the family Pelagibacteraceae ($z$ score = −1.94) and genus *Prochlorococcus* ($z$ score = −1.25), which hold some of the most abundant organisms on Earth[52,53]. These taxa, known for their highly streamlined genomes[54], are found in aquatic samples with a uniform microbial composition (Supplementary Fig. 5b) and thus have a low SNB. While the family Pelagibacteraceae contains both marine and freshwater representatives (in the SAR11 and LD12 clades, respectively)[55], in our dataset it

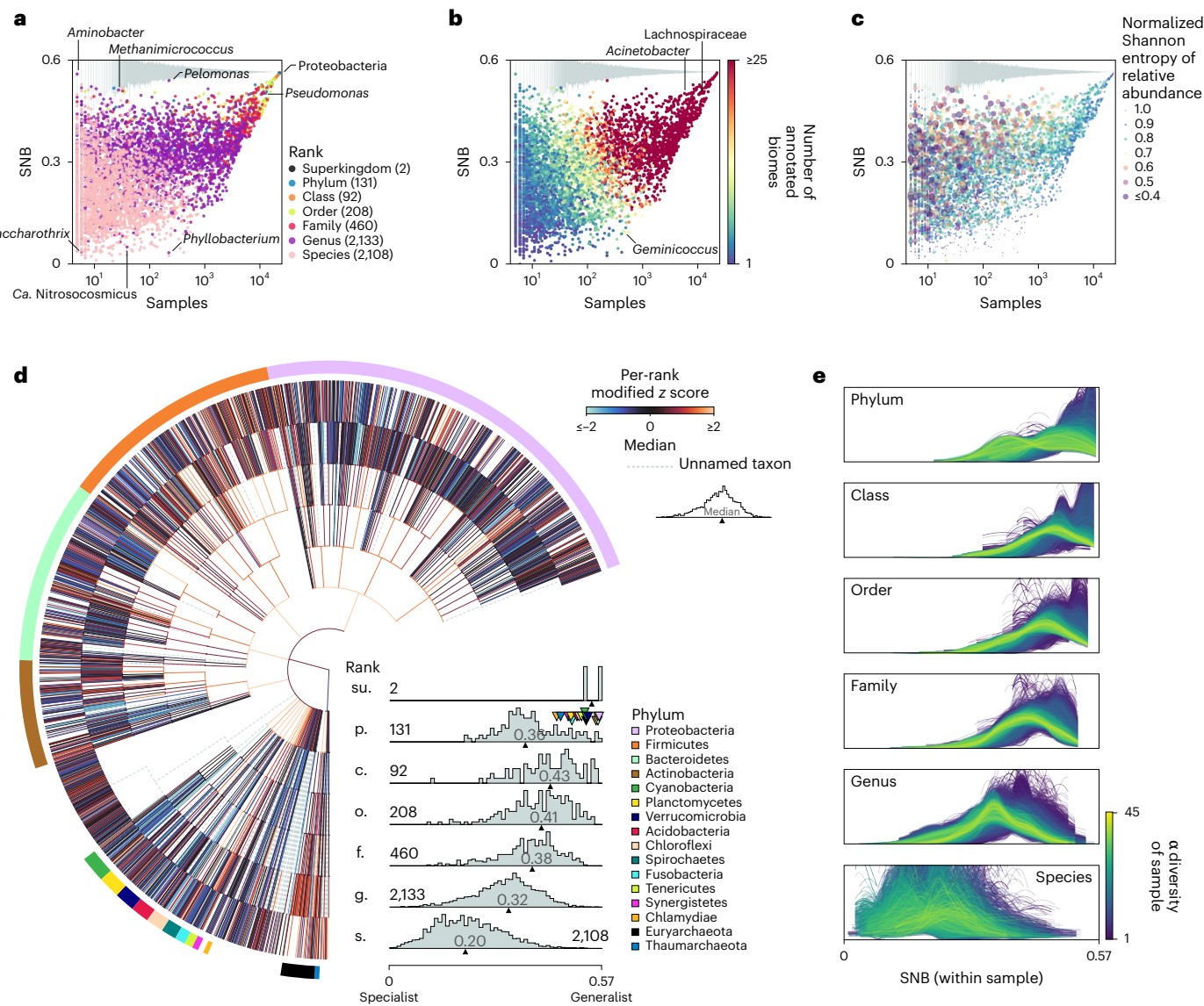

**Fig. 2 | SNB throughout the prokaryotic tree of life and across samples.**
**a**–**c**, Relationship between SNB and the number of samples in which a taxon is found, coloured by rank (**a**), number of annotated biomes (**b**) and normalized Shannon entropy of relative abundance (**c**). The grey bars on top show the range of SNB scores of imaginary taxa that were present in 100 randomly picked subsets of samples of the specific size. The locations of some outlier taxa are indicated. Numbers within brackets indicate the number of taxa measured at each rank. The normalized Shannon entropy of relative abundance across samples is represented as shading for each taxon in **c**. Both colour coding and the size of the markers represent the Shannon entropy. Note that higher entropy is indicated with smaller markers. Relative abundance across samples was more constant for social specialists than for social generalists. Ca., *Candidatus*. **d**, SNB of taxa throughout the prokaryotic tree of life. SNB scores are standardized per rank based on the median absolute deviation (modified *z* scores), with low *z* scores representing taxa that are relatively specialist for their rank and high *z* scores representing taxa that are relatively generalist. The distributions of SNBs at different taxonomic ranks are shown as histograms, for which the numbers on the distributions show the number of taxa at that rank. The most diverse phyla are colour coded. **e**, Distribution of SNBs within samples at different taxonomic ranks. The α diversity of a sample was calculated on the rank order.

is found primarily in marine samples (Supplementary Fig. 5b). This highlights that future sampling of even more habitats, combined with more sensitive detection methods, could change or refine SNB scores for some taxa. The genus *Roseobacter*, whose members are considered marine metabolic generalists with large genomes and a versatile metabolism[56,57], is found in more diverse samples (Supplementary Fig. 5b) and has an SNB closer to the median of all genera (*z* score = −0.30). At the generalist end of the spectrum are taxa that are ubiquitously present in our dataset (Fig. 2a,b), such as the genera *Acinetobacter* (*z* score = 2.30) and *Pseudomonas* (*z* score = 2.33; however, this genus may be ubiquitous in part because it is a common contaminant of

DNA extraction kits[58]) (Supplementary Fig. 5b). The family Lachnospiraceae (*z* score = 1.74; found in over half of all samples; *n* = 11,887) and its genera (median *z* score = 0.79)—obligate anaerobes that were previously regarded as habitat specialists[27]—also have a high SNB for their ranks, highlighting the heterogeneity of the communities in which they are found.

**Generalists dominate, whereas specialists are stable but scarce**
Next, we set out to find patterns in SNB. We focused our analysis on genera because they balance a high taxonomic resolution with a good representation in the dataset (Fig. 1e) and show a broad range of SNB

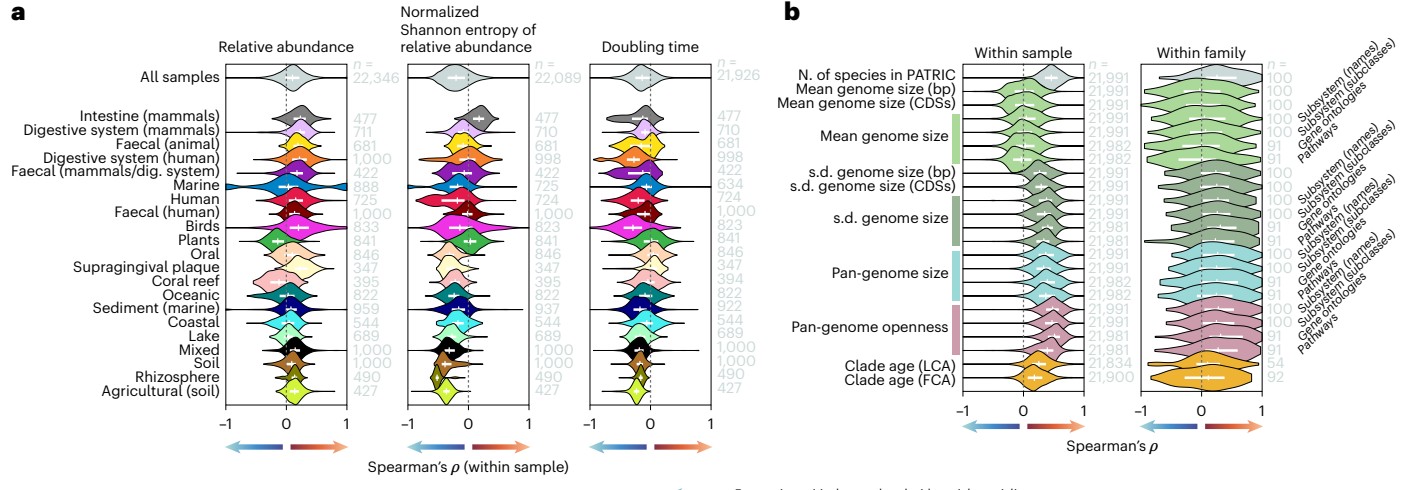

**Fig. 3 | Ecological and genomic features correlated with SNB. a**, Spearman's rank correlation coefficient ($\rho$) within samples between SNB and features related to local dominance on the rank genus. The violin plots depict the distribution of $\rho$ across all samples or those from the annotated biomes with the most samples. Annotated biomes are arranged according to mean α diversity. Positive values indicate that the feature is positively correlated with social generalists and negative values indicate that the feature is positively correlated with social specialists. Doubling time estimates are from the EGGO database. dig., digestive. **b**, $\rho$ between SNB and genomic features in the PATRIC database on the rank genus within samples and within families. The violin plots depict the distribution of $\rho$ across all samples and across all families with at least five genera. Genome sizes are given in base pairs (bp) and numbers of coding sequences (CDS). Genomic measures with annotations to the right are in numbers of unique functions for that specific functional universe. Genome size estimates for a genus are based on the genome size of its species, which is defined as the mean size of all strains for base pair and coding sequence measures, and as the majority set of functions of all strains for the functional universe measures. Pan-genome openness is the total pan-genome size divided by the mean genome size. Correlations with time to the last common ancestor (LCA) and first common ancestor (FCA) are based on the TimeTree database. Numbers to the right of violins show sample sizes. Lines within violin plots show the interquartile range and median. Supporting data are available in Supplementary Data 6 and 7. N., number; s.d., standard deviation.

values (Fig. 2d,e), allowing for a comprehensive investigation of niche range strategies.

It has been suggested that generalists, being Jacks of all trades, can be masters of none[11], while specialists are adapted to become dominant within their habitats under stable conditions[59]. The niche range may thus reflect a trade-off, where specialists gain local dominance at the expense of ecological versatility. Alternatively, computational models of microbial metabolism have suggested that metabolically flexible generalists have faster growth rates than specialists[60]. We correlated SNB with local abundance and found that social generalists are dominant in most annotated biomes, as indicated by a consistent positive correlation across samples, with exceptions including marine host organisms such as corals, seagrasses and macroalgae (Fig. 3a). SNB positively correlates with abundance within samples, meaning that social generalists locally outcompete their more specialist neighbours, disputing the expected trade-off mentioned above. While these are general results based on correlations across samples, an exception is *Prochlorococcus*, which has a low SNB but a high local abundance (mean relative abundance = 0.63%). This genus is among the top 10% of the most abundant genera (Supplementary Data 3) and in the majority of its samples belongs to the top 20% of genera in terms of local abundance. Local dominance of habitat generalists has previously been observed in specific environmental settings such as highly dynamic sandy ecosystems[61]. Some soil microorganisms are both abundant and ubiquitous[62] and only ~500 dominant phylotypes (that is, 2%) represent >40% of soil bacterial communities[63]. Our results show that these observations reflect a general pattern wherein generalists are dominant.

Whereas samples are typically dominated by social generalists, we find that the relative abundance of generalists is more variable across samples than that of specialists, whose abundance is relatively stable. This is evident when comparing the niche range of organisms that locally co-occur within samples, where social generalists have a higher variability of relative abundance than social specialists (Fig. 3a). It is also evident for taxa throughout the prokaryotic tree of life, where social specialists have an even relative abundance while taxa with a high variability of relative abundance are social generalists (Fig. 2c). Even *Prochlorococcus*, while having a high local dominance for a social specialist, still has an even abundance across samples (normalized Shannon entropy of relative abundance = 0.86).

Our data counter the classic Jack of all trades argument, which suggests that specialists should have a local fitness advantage at the expense of ecological versatility. We explored possible explanations for the local dominance of generalists over specialists and their relatively variable abundance. First, although generalist genera contained more species than specialist genera in the total dataset (Supplementary Fig. 18a), we did not find evidence that they also contain a higher number of species within samples (Supplementary Fig. 18a), but note that only a small fraction of genera could be taxonomically classified on the species rank (Supplementary Fig. 18b). Alternatively, SNB may reflect the classical distinction between r strategists and K strategists[64]. Social specialists have a low but constant abundance near carrying capacity (K selected) and some (but not all) social generalists are opportunistic taxa that reach high relative abundance when circumstance permits (r selected). To test this hypothesis, we compared the SNBs of microorganisms with their predicted maximal growth rates based on the EGGO database[65] (Fig. 3a) and confirmed that, within samples, social generalists have shorter doubling times than social specialists. These results support the idea that generalist genera include more opportunistic growers than specialist genera.

### SNB reflects genomic heterogeneity
Next, we used our dataset to assess the suggestion that social generalists have large genomes that encode many functions, reflecting a versatile metabolism that allows them to colonize diverse habitats[25,27]. For example, bacteria that are found in a diverse range of habitats encode more extracellular proteins than bacteria that are restricted to few habitats[26],

and habitats with temporal variation may select for larger genomes[66]. In contrast, specialization may be associated with a reduction in genome size due to a loss of unnecessary genes (as has been observed in members of the phylum Planctomycetes transitioning from soil to freshwater habitats[67]) or genome streamlining[68] (which is common in oligotrophic marine waters[69,70]). Genomic versatility of high-ranking taxa, reflected in a large pan-genome[71,72], may either result from small yet diverse genomes in individual subtaxa (open pan-genome) or genomically versatile yet functionally similar strains (closed pan-genome). We set out to identify genomic features associated with SNB using publicly accessible genome sequences from the Pathosystems Resource Integration Center (PATRIC) database[73] (Supplementary Data 5). These features include the mean genome size of all species in the genus, the variation in these genome sizes, the pan-genome size (that is, the total number of functions present in all genomes) and the pan-genome openness (calculated as the pan-genome size divided by the mean genome size). The PATRIC database contains genome sequences for 1,704 of the 2,133 genera that we investigated in our global microbiome dataset (Supplementary Data 3). Although these genomes probably belong to different strains or species than those observed in MGnify, we decrease the inconsistencies in our analysis by assessing their genomic features at the genus rank.

We compared genera within samples (Fig. 2e) for an ecological view and within their taxonomic families for an evolutionary view. Both perspectives gave qualitatively similar results (Fig. 3b), indicating that genomic signatures of SNB (see below) are generalizable across habitat and phylogeny. Although the number of samples is larger than the number of families, the correlation between genomic features and SNB is more consistent within samples than within families, possibly suggesting that ecology is a stronger driver of (pan)-genome evolution than phylogenetic history[74].

When comparing taxa across all samples, we found no consistent correlation between SNB and genome size, whether measured in the number of nucleotides, genes or unique functions (Fig. 3b). We did, however, observe that the genomes in generalist genera are more variable in size than the genomes in specialist genera, as seen in their standard deviation. Moreover, the pan-genomes of generalist genera contain more functions, in line with theoretical models that suggest that the ability to migrate to new niches is associated with pan-genome size[75]. The pan-genome size of microorganisms may be positively associated with effective population size[76], which may be larger for social generalists. The same study found that rapidly growing microorganisms have large effective population sizes, in line with our earlier discussed observation that opportunistic growers are generalists. Finally, the pan-genomes of social generalists are more open than those of social specialists. These results did not depend on the higher number of species in generalist genera (Supplementary Fig. 19).

In conclusion, species in specialist genera are genomically more similar than species in generalist genera, which is reflected in more similar genome sizes and less variation in functions, and an associated smaller and more closed pan-genome than generalist genera. We hypothesize that the observed genomic flexibility allows members of generalist genera to rapidly acquire the genes needed to thrive in a given local environment[77,78], with their higher growth rate potential allowing them to outgrow specialists. The correspondence between genomic heterogeneity and the heterogeneity of communities (SNB) confirms the strong association between ecological and genomic diversification. To further explore this diversification we correlated SNB with clade age based on the TimeTree database[79]. It was previously suggested that habitat generalist species are evolutionarily younger than habitat specialist species[27]. Our data do not allow analysis of these trends at the species rank, but at the genus rank we found that social specialists are younger than generalists, as indicated by a consistent positive correlation between SNB and clade age (Fig. 3b). Together, these results support a model of continuous diversification whereby

old generalist clades that share a diverse pan-genome may invade new niches, leading to the emergence of specialized subtaxa.

## Two contrasting genomic niche range strategies
As discussed above, SNBs of genera were not consistently associated with mean genome size. However, we did observe a habitat-dependent relationship between genome size and social niche range (Fig. 4a). We found two contrasting strategies that broadly depended on the α diversity of the samples. In samples with low local diversity (α diversity ≤ 11; Fig. 4b), including most animal-associated and saline habitats, there was a mean positive correlation between genome size and SNB. In contrast, in samples with high local diversity (α diversity > 11; Fig. 4b), including most free-living non-saline habitats and the rhizosphere, the correlation was often negative. Because databases contain a majority of samples from animal-associated and marine habitats with relatively low α diversity (Fig. 4c), and because genome size estimates are often based on cultivated microorganisms that differ markedly from environmentally derived genomes[80], previous suggestions of a positive correlation between genome size and niche range may have overlooked the contrasting correlations.

In habitats with low α diversity, social generalists tend to have large genomes that may encode the functions needed to utilize many different resources, whereas social specialists (low-diversity specialists) have the smallest genomes known (Fig. 4d,e). The coding density—a signature of genomic streamlining[68]—is significantly lower in low-diversity specialists than in social generalists ($P < 0.003$; one-tailed $t$-test; measured as the number of coding sequences per base pair, with generalists defined as SNB > 0.35 (697 genera) and low-diversity specialists defined as present in samples with a mean α diversity ≤ 11 (552 genera)). This suggests that genome streamlining is not the common route to genome reduction in low-diversity specialists. Instead, their small genomes could reflect specialization to habitat-specific metabolites and the loss of genes through drift[67]. In addition, cooperating metabolic specialists could supplement each other's nutrient requirements[50] or they could depend on the co-occurring generalists.

Although the genomes of social generalists are large compared with those of low-diversity specialists (Fig. 4d), the genomes in samples with low α diversity are still moderate in size compared with the large genomes in samples with high α diversity (Fig. 4e). Samples with high α diversity contain many specialists (Fig. 2e) and their high richness may be a driver of specialization through competitive exclusion, as has been suggested to explain a negative correlation between niche breadth and local diversity in eukaryotes[81]. There is negative correlation between genome size and social niche range in samples of high α diversity, where social specialists (high-diversity specialists) have larger genomes than co-occurring social generalists (Fig. 4b). The largest known prokaryotic genomes belong to high-diversity specialists (Fig. 4d); for example, the genus *Polyangium*, with a mean genome size of 12.7 megabases and an SNB of 0.21 ($z$ score = −1.36). Selection may favour large genomes in habitats with diverse but scarce nutrient availability where slow growth is no disadvantage, such as in soils[82,83]. Moreover, in contrast with the earlier-mentioned cooperative taxa, microorganisms in competitive consortia carry many metabolic functions[50]. High-diversity specialists may thus reflect a competitive metabolism. Social generalists in these habitats may use metabolites that are irregularly available and rapidly depleted, consistent with their opportunistic nature and variable occurrence. Alternatively, they could exploit metabolic byproducts generated by metabolic specialists[19]. Regardless of the mechanism, it appears that adaptation of high-diversity specialists to their habitats by genome expansion decreases their competitiveness in differing communities.

## The pan-genomes of social specialists and generalists
To further characterize generalist and specialist microorganisms, we explored differences in genomic content by dividing all genera into

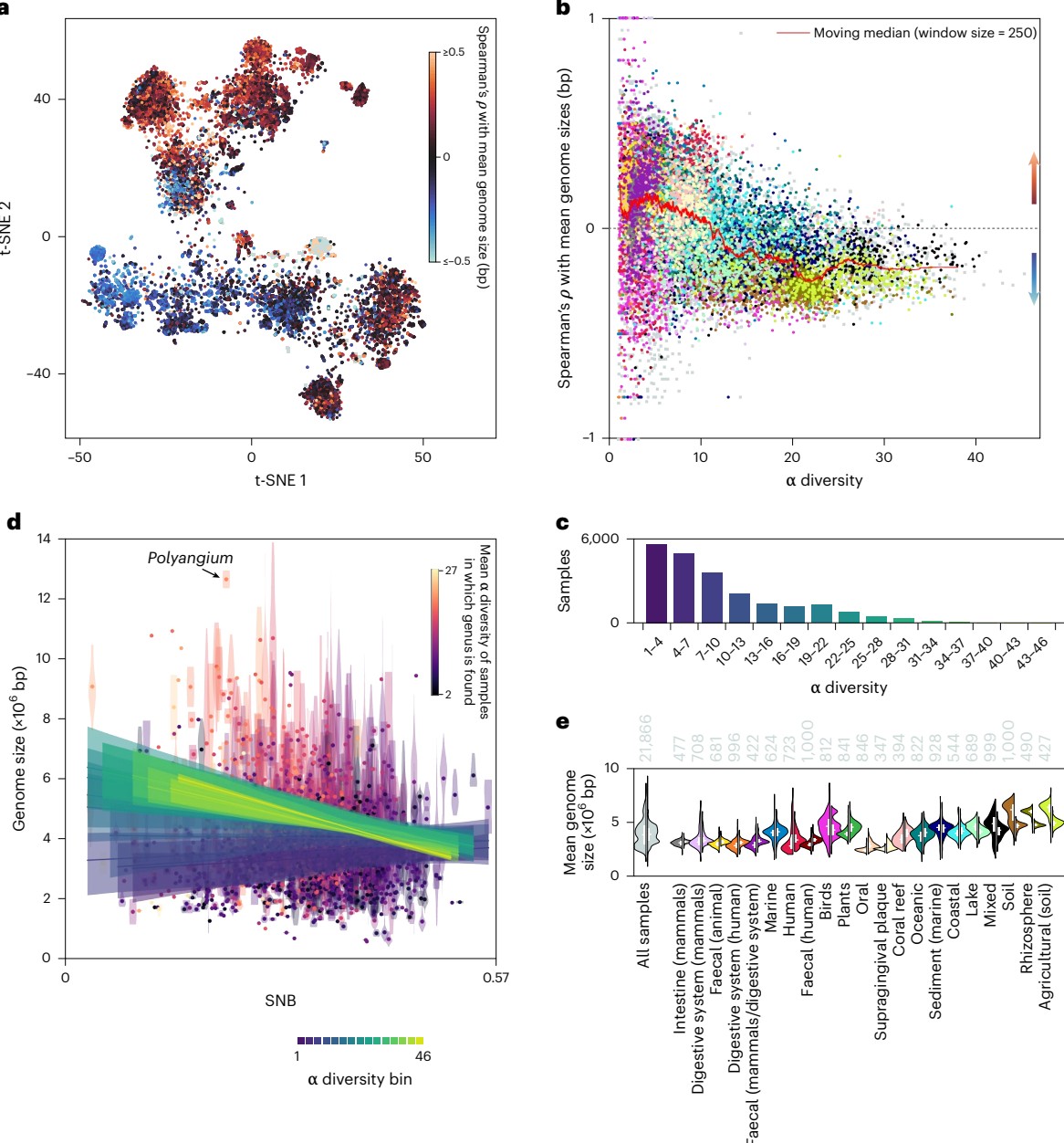

**Fig. 4 | Contrasting genomic niche range strategies. a**, Spearman's rank correlation coefficient ($\rho$) per sample between SNB and mean genome size on the rank genus plotted on the t-SNE (Fig. 1d). Positive values indicate an association with social generalists and negative values indicate an association with social specialists. **b**, $\rho$ as a function of the α diversity of the sample. The colour coding represents annotated biomes (see **e**). **c**, Number of samples per bin of α diversity. **d**, SNB versus genome size on the rank genus. The violin plots show the distribution of genome sizes of species within a genus and the dots represent mean values. The lines depict the mean of linear regression lines between SNB and mean genome size of all samples in a specific bin of α diversity. The shaded areas show the interquartile range of the regression lines. **e**, Violin plots depicting the distribution of mean genome size of the top 25% social specialist taxa (left) and top 25% social generalist taxa (right) within a sample across all samples or those from the annotated biomes with the most samples. The annotated biomes are arranged according to mean α diversity. The numbers at the top of the violin plots show the sample size. The lines within the violin plots show the interquartile range and median. Supporting data relating to **a** and **b** are available in Supplementary Data 6.

two groups based on SNB (Fig. 5a) and performing gene set enrichment analysis (GSEA)[84] on the genus-level pan-genomes. We performed two GSEAs: one with all genera comparing social specialists with social generalists (Fig. 5b and Supplementary Data 8); and one comparing low-diversity specialists with high-diversity specialists (Fig. 5c and Supplementary Data 9).

The functions enriched in social generalists (false discovery rate (FDR) < 0.1; Fig. 5b) included associations with genome fluidity, such as (pro-)phages and plasmid-related functions, highlighting the

mechanisms by which they keep an open pan-genome. Other generalist functions reflected an investment in species–species interactions, observation and response to a fluctuating environment. Of the 33 generalist-enriched functions, 13 were related to metabolism, including functions associated with secondary metabolites, such as coenzyme $F_{420}$ (refs. 25,85). We also found quorum sensing and biofilm formation, adhesion, locomotion via the flagellum, and functions concerning the cell envelope and transport across it (S-layers, protein secretion systems and siderophores) to be generalist enriched. Finally,

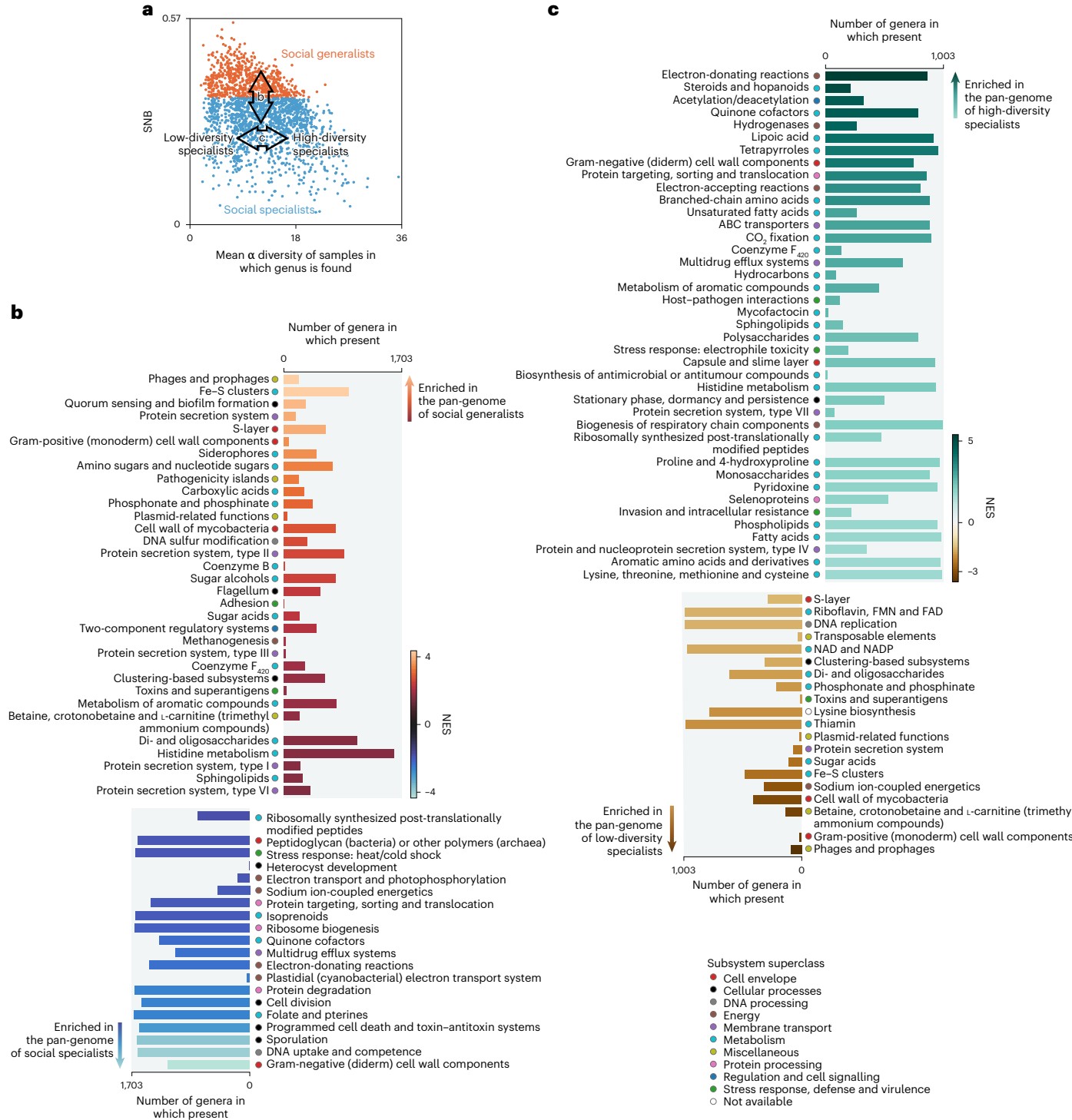

**Fig. 5 | Functional characterization of generalists and low- and high-diversity specialists. a**, Division of genera in social specialists (SNB < 0.35) and social generalists (other). Arrows represent the direction of the GSEA in panels **b** and **c**. **b**, GSEA on the pan-genome of all genera sorted by SNB, on the level of subsystem subclasses. **c**, GSEA on the pan-genome of specialist genera sorted by mean α diversity of the samples in which they are found, on the level of subsystem subclasses. Enriched functions (FDR < 0.1) in **b** and **c** are sorted according to normalized enrichment score (NES). The size of the bars indicates the total number of genera in the dataset having the function. The most enriched functions are in the upper and lower parts of the panels. Full GSEA information, including analyses of other functional universes (subsystem names, gene ontologies and pathways), is available in Supplementary Data 8 and 9.

pathogenicity islands could point to opportunistic interactions with eukaryotic host organisms.

Fewer genomic functions were enriched in social specialists than in social generalists (Fig. 5b). Specialist-enriched functions include energy-related processes and some specifically related to

Cyanobacteria, such as heterocyte formation, which is involved in nitrogen fixation in one evolutionary lineage of the phylum[86]. Most functions that were enriched in specialists also occurred in the pan-genomes of many generalist genera (Fig. 5b), suggesting that the smaller pan-genome size of social specialists does not involve consistent loss

of functions. The absence of widespread specialist functions highlights that there are many ways to be a specialist. There is not a single type of social specialist, but instead many different specialists exist, each with a functional arsenal that fits its niche.

Comparing low-diversity specialists with high-diversity specialists (Fig. 5c), we observed several specific metabolic adaptations to these different types of habitat. Half of the 60 enriched functions in the GSEA were related to metabolism. High-diversity specialists have more enriched functions than low-diversity specialists. For example, functions associated with stationary phase, dormancy and persistence are enriched in high-diversity specialists, consistent with slow growth and persistence in soil. Moreover, functions related to lipid metabolism (for example, steroids and hopanoids, (unsaturated) fatty acids, sphingolipids and phosopholipids) are enriched in high-diversity specialists. Low-diversity specialists, like social generalists, also contain some functions associated with genome fluidity (for example, transposable elements, (pro-)phages and plasmid-related functions), suggesting that their genomes, although small in size, may still be in flux.

## Discussion

We present an SNB score for microbial taxa that is based on the community similarity of the samples in which they occur. Integrating information from over 22,000 samples, SNB represents a global and comprehensive view on niche range throughout the prokaryotic tree of life. With continued and ever-deeper sequencing efforts and associated expansion of public databases, the environmental and taxonomic resolution of our picture of the microbial world increases, as does our understanding of the processes shaping microbial niche breadth. In contrast with earlier suggestions, we found that most habitats are dominated by generalists. Specialists occur at low but stable abundances.

Generalist genera are older than specialist genera and have large and open pan-genomes with which they have adapted to different habitats. Individual genome size and SNB are differentially related depending on the diversity of the habitat, with social generalists having larger genomes than social specialists in low-diversity habitats and smaller genomes than social specialists in high-diversity habitats. High-diversity specialists may need a large genetic repertoire as they are continually exposed to many different interaction partners and possibly high environmental variability at small spatial scales. Low-diversity specialists have decreased genome sizes due to the loss of unnecessary functions. Large genomes may thus reflect increased environmental versatility in two different settings. In habitats with low local diversity, generalists are relatively versatile, as they can survive in a range of different communities. In habitats with high local diversity, specialists are relatively versatile, allowing them to persist in their local complex community. Since social generalists and social specialists are dispersed throughout the prokaryotic tree of life, these genomic adaptations have repeatedly occurred and represent fundamental eco-evolutionary processes.

## Methods

### Sample selection

We downloaded taxonomic profiles deposited in the MGnify microbiome resource[37] on 20 August 2019. MGnify contains taxonomic profiles based on studies that amplify taxonomic marker gene regions (amplicons), shotgun metagenomics studies and shotgun metatranscriptomics studies. We selected taxonomic profiles that were constructed with the pipeline 4.1 version of MGnify and based on the small subunit ribosomal RNA gene, contained at least 50,000 taxonomically annotated reads at the rank superkingdom and had <10% of those reads classified as eukaryotic. We randomly picked one taxonomic profile per sample in cases where there were multiple. To balance the large over-representation of several environments in the database (for example, human gut, soil and ocean), at most 1,000 samples were randomly selected per annotated biome. The 22,518 selected samples (Supplementary Data 1) spanned 140 different annotated biomes across a wide geographical range and consisted of amplicon, metagenomic, metatranscriptomic and unknown experiment types (Supplementary Fig. 1).

We removed eukaryotic classifications including those classified as mitochondria and chloroplast from the taxonomic profiles, as well as those not classified at the taxonomic rank superkingdom. When relative abundances were used, they were calculated as the number of reads assigned to a taxon divided by the total number of prokaryotic reads, unless otherwise stated in the section 'Ecological dissimilarity measures'.

### Ecological dissimilarity measures

We calculated ecological dissimilarity between all sample pairs based on their taxonomic profiles (compositional dissimilarity) at different taxonomic ranks using ten commonly used ecological measures: Aitchison distance; Bray–Curtis dissimilarity; Sørensen–Dice coefficient; Jaccard distance; weighted Jaccard distance; Kendall's $\tau_b$ coefficient; Pearson correlation coefficient; Spearman's rank correlation coefficient; unweighted UniFrac distance; and weighted UniFrac distance. Some are true distance or dissimilarity measures, whereas others can be readily converted to a scale from 0–1, with 0 being compositionally similar. The three correlation measures were converted to dissimilarity with the formula $0.5 - (\text{coefficient}/2)$ and we used $1 - \text{Sørensen–Dice}$ coefficient.

Taxa that were represented by fewer than five reads in the sample were removed before dissimilarity calculations. This was done per rank; therefore, the total number of included reads for a sample could differ depending on the rank considered. To ensure that the pairwise calculations were based on the deepest attainable resolution, we decided on a low absolute read cut-off as opposed to a relative abundance cut-off. For each pairwise calculation, we only included taxa that were present in the union of the two samples, thus avoiding the vast scarcity (that is, the presence of zeros in the abundance matrix) often associated with microbiome studies. This scarcity is especially likely because our study compares many different habitats. Those taxa that were only present in one of the samples were given an abundance of zero in the other for all ecological dissimilarity measures except the Aitchison distance, which cannot handle zeros. For the Aitchison distance, a pseudocount was added. This pseudocount differed per pair of samples and was based on the lowest relative abundance that could be reached by an undetected taxon (namely, one read in the sample with the highest number of taxonomically annotated reads). We defined $N_1$ as the sum of reads represented by the taxa in sample 1 and $N_2$ as the sum of reads represented by the taxa in sample 2, with $N_1 \geq N_2$. A pseudocount of one read was added to all taxa in sample 1 and a pseudocount of $1/N_1 \times N_2$ reads was added in sample 2.

We calculated the ecological dissimilarity measures at all ranks up to phylum with three different methods for dealing with unknowns in the data. For the UniFrac distances, we used a different method (see below). For approach (1), we considered any taxon on the specific rank. If there was no classification at that rank but the taxon contained lower rank classifications, the first classified rank below was used. If there was no classification at the specific rank and no lower rank classification, we used the first classified rank above. For approach (2), we exclusively considered taxa that were classified at the specific rank. Taxa that were classified at lower or higher ranks alone were removed. For approach (3), we treated taxa that were not classified at the specific rank but did have lower rank classifications the same as in approach (1). If taxa had no classification at the specific rank or at a lower rank, we used the first classified rank above, unless the taxon was present in both samples. In this case, the taxon was removed. The rationale is that for these taxa it is unknown whether they are the same or different for the rank of interest.

UniFrac distance takes relatedness between taxa into account. We used distance across the taxonomic tree as a measure for relatedness, with the distance between successive ranks defined as 1. We used the

EMDUnifrac implementation[87], which is suited for samples with many unknowns because it allows for the placement of taxa at different ranks in the tree. UniFrac distances were calculated at the ranks species, family and class. For taxa that had no classification at the specific rank but did have a lower rank classification, we used an artificial classification based on the first classified rank below, ensuring uniqueness of the taxon and appropriate distance to the root. Taxa that did not have a classification at the specific rank or a lower rank were placed at the first classified rank above in the tree.

For the ecological dissimilarity calculations that took the number of reads into account, the numbers of reads per taxon were converted to relative abundance values by dividing by the sum of reads represented by the taxa in the sample (for example, for Bray–Curtis dissimilarity calculations and the addition of pseudocounts before Aitchison distance calculations). As explained above, the taxa considered in a sample may have differed per method of dealing with unknowns, and so may the relative abundance of a taxon.

Because the taxonomic profiles contain many unknowns at lower ranks, pairwise comparisons are sometimes based on only few taxa. For each rank and method of dealing with unknowns, samples were removed that did not contain any taxon at that rank. If the pairwise comparison was based on one taxon, we set the dissimilarity to 0. We removed samples from the correlation measures whose correlation coefficient with itself could not be calculated.

### Permutatational multivariate analysis of variance

Permutatational multivariate analysis of variance pseudo $F$ statistics were calculated for all ecological dissimilarity measures with the scikit-bio version 0.5.5 implementation (http://scikit-bio.org/). As predefined groups, we used either the annotated biomes of the samples or their experiment types. $P$ values were based on 99 random permutations and we calculated the coefficient of determination ($R^2$) with the formula:

$$R^2 = \frac{F}{F + \frac{N-G}{G-1}}$$

were $F$ is the pseudo $F$ statistic, $N$ is the sample size and $G$ is the number of groups.

### Diversity measures

Diversity measures were calculated for the subset of samples belonging to an annotated biome and in which a taxon was found. If a subset contained fewer than three samples, it was excluded from analysis. Taxa were removed whose relative abundance was less than 1/10,000. We used approach (2) to deal with unknowns, as explained in the section 'Ecological dissimilarity measures'.

Zeroth-order α diversity (that is, richness—the mean number of taxa found in a set of samples) was calculated for all ranks. Zeroth-, first- and second-order α diversity ($^qD_\alpha$) and β diversity ($^qD_\beta$) were calculated on the taxonomic rank order and based on relative abundances, with $^qD_\beta$ defined as the total effective number of taxa ($^qD_\gamma$) divided by $^qD_\alpha$. $^qD_\gamma$ was calculated based on the summed relative abundance of the individual samples. For the first- and second-order diversity measures, two samples were excluded that did not contain any classification at order rank after the relative abundance threshold.

When the terms α and β diversity are used, we refer to first-order diversity measures on the rank order unless otherwise stated. For a more in-depth discussion of these diversity measures, see ref. [88]. The Shannon entropy and Gini–Simpson index, which were used for diversity calculations, were calculated with the scikit-bio version 0.5.5 implementation.

### Local dominance and Shannon entropy across samples

For each taxon, we calculated local dominance and Shannon entropy. Local dominance was defined as the mean relative abundance across all samples in which the taxon was found. Shannon entropy (base $e$) was used as a measure for the randomness of its relative abundances across these samples ($N$) and was normalized by dividing by ln[$N$].

### SNB definition

SNB was defined as the mean of the pairwise dissimilarity between the samples in which a taxon was found, $n$:

$$SNB = \frac{\sum_{i=1}^{n} \sum_{j=1}^{n} d_{ij}, \ i \neq j}{n^2 - n}$$

with the dissimilarity $d_{ij}$ based on the Spearman's rank correlation coefficient ($0.5 - (\rho/2)$) on the rank order with method (2) for dealing with unknowns (see the section 'Ecological dissimilarity measures'). A taxon was considered present in a sample if it had a relative abundance of at least $1 \times 10^{-4}$. A taxon with a low score was thus found in samples with similar taxonomic profiles (social specialists) and a taxon with a high score was found in dissimilar samples (social generalists). Taxa that were present in fewer than five samples were removed from analyses unless otherwise stated.

To benchmark SNB, we also calculated SNB with different detection thresholds of $1 \times 10^{-3}$ and $1 \times 10^{-5}$. SNB was moreover calculated for imaginary taxa (iSNB) that were present in all samples from a given annotated biome, in half of the samples from a given annotated biome (100 random permutations per annotated biome) and in all samples from pairs of annotated biomes. In addition, iSNB was calculated for randomly picked sets of samples of equal size to the encountered taxa (100 random permutations per sample size). Lastly, we calculated SNB for real taxa only based on the marine and human hierarchical subsets of the samples.

### Selection of genomes

We downloaded all genomes from the PATRIC genome database[73] that had a quality marked as good and were not plasmids on 14 November 2019. We only included genomes for which we had a valid taxonomy ID in our NCBI taxonomy[89] files that were downloaded on the same date. PATRIC contains identical genomes with different identifiers. We identified replicate genomes based on concatenated DNA sequences and concatenated sorted DNA sequences and removed all but one. In cases where identical genomes had different taxonomic annotations, all were discarded. Completeness and contamination estimates were generated with CheckM version 1.0.7 (ref. [90]) in the lineage-specific workflow. We excluded genomes for which completeness − 5 × contamination < 70. The final selection consisted of 225,101 prokaryotic genomes representing 34,304 species, from both cultures and environmental sequencing projects (Supplementary Data 5).

Inferences about (pan)-genome size and genomic functions (see below) and number of subtaxa were made at all taxonomic ranks and reconstructions at higher ranks were based on lower-rank taxa in the PATRIC database that were not always present in the MGnify dataset.

### Genome size estimates and GC content

Genome size (in number of base pairs and number of coding sequences) and GC content were obtained from the metadata in the PATRIC database. We corrected genome size estimates by taking completeness and contamination into account, via multiplication with a scaling factor $s$:

$$s = \frac{100}{completeness + contamination}$$

For each measure including the number of coding sequences per million base pairs, we reconstructed species by averaging the values of its genomes. For higher ranks, mean values were calculated for all species belonging to the taxon. Some high-ranking taxa contain many low-ranking taxa from the same taxonomic group, such as genus or family. To correct for this over-representation and possible skew towards

the values of these taxa, we also calculated mean values by averaging over the taxonomy at all ranks (taxonomy-corrected values). For example, the size of a family is the mean of the sizes of its genera and the size of a genus is the mean of the sizes of its species. These values were calculated for the ranks family and higher and are available in Supplementary Data 5.

### Genome functions

Functional profiles of the genomes were created based on the PATRIC annotations of coding sequences for three functional universes: subsystems, gene ontologies and pathways. For gene ontologies, the profiles were based on the exact terms found in the annotation files, whereas for subsystems we made different profiles for the name and subclass level of the hierarchy. Genomes with ≤20 unique functions were discarded from further analyses for subsystem names, gene ontologies and pathways. For the included list of genomes in each analysis, see Supplementary Data 5.

Functional genome size was defined as the number of unique functions present in a species. A function was considered present in a species if at least 50% of the genomes with this species annotation contained it in the PATRIC database. Mean functional genome sizes were calculated for all taxa, as well as the standard deviation. Pan-genomes were defined at all ranks as the total set of unique functions present in the genomes of a taxon.

Pan-genomes can be open or closed, meaning that they can be more or less susceptible to changes in gene content[71,72]. We devised a score that represents pan-genome openness for all ranks higher than species. Pan-genome openness was defined as the total pan-genome size divided by the mean pan-genome size of a species. Because taxa with many subtaxa tend to have large pan-genomes, we also calculated pan-genome features for a random subset of three daughter species (1,000 random permutations per taxon) to correct for this effect of taxonomy. This measure thus reflects how many functions are on average added to the pan-genome by including two more species. Permutated measures were calculated for all taxa with at least three species.

### GSEA of pan-genomes

To detect functions that were significantly enriched in social specialists and generalists, we deployed GSEA[84] based on the pan-genomes of genera. We performed a GSEA on all genera sorted by SNB to compare specialists with generalists, and on specialist genera (SNB < 0.35) sorted by α diversity to compare low-diversity specialists with high-diversity specialists. We used the classical Kolmogorov–Smirnov statistic for the enrichment score ($p = 0$). Enrichment score normalizations and $P$ values were based on 100,000 random permutations of the gene set. Multiple hypothesis correction was carried out via the FDR as suggested in ref. 84. GSEA computations were done with a modified version of the algorithm.py script from GSEApy version 0.7.3.

### Growth rate and clade age estimates

We downloaded the maximal growth rate predictions of RefSeq genomes from the EGGO database[65] and defined species- and genus-rank maximal growth rates as the mean of their genomes.

Clade ages were based on the TimeTree database[79]. Times to the first and last common ancestor were extracted from the species rank phylogenies of bacteria and archaea using ete3 version 3.1.1 (ref. 91),

### Software packages used for calculations and visualizations

Calculations were done with the Python 3 standard library, NumPy[92] and the SciPy library[93] unless otherwise stated. Visualizations were done with Python 3 and Matplotlib[94] in JupyterLab (https://jupyter.org/), with the use of NumPy, pandas[95] and seaborn (https://seaborn.pydata.org/). Principal coordinates analysis (PCoA) was performed with the scikit-bio version 0.5.5 implementation. t-SNE was performed with scikit-learn version 0.21.3 (ref. 96). Samples were drawn on the world map with Cartopy version 0.17.0 (http://scitools.org.uk/cartopy/) using Natural Earth data (https://www.naturalearthdata.com). The taxonomic tree was visualized with iTOL[97] and the hierarchical tree of annotated biomes was visualized with ete3 version 3.1.1 (ref. 91).

### Reporting summary

Further information on research design is available in the Nature Portfolio Reporting Summary linked to this article.

## Data availability

All of the data analysed during this study are included in this article and Supplementary Data 1–9 or available in public repositories. The selected samples from the MGnify resource (https://www.ebi.ac.uk/metagenomics/) are described in Supplementary Data 1. The cleaned taxonomic profiles based on these data are available in Supplementary Data 2. The selected genomes from the PATRIC database (https://www.bv-brc.org/) are described in Supplementary Data 5. Measures derived from the PATRIC genomes and the EGGO (https://github.com/jlw-ecoevo/eggo) and TimeTree (https://timetree.org/) databases are available in Supplementary Data 3.

## Code availability

All of the code used for this manuscript is available from Zenodo at https://doi.org/10.5281/zenodo.7651594. A stand-alone script to calculate SNB is available from https://github.com/MGXlab/social_niche_breadth.

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

## Acknowledgements

We thank J. K. van Amerongen for technical support. This work is funded/supported by the European Research Council (Consolidator Grant 865694: DiversiPHI to B.E.D.), the Deutsche Forschungsgemeinschaft under Germany's Excellence Strategy (EXC 2051; Project-ID 390713860 to B.E.D.) and the Alexander von Humboldt Foundation in the context of an Alexander von Humboldt Professorship founded by the German Federal Ministry of Education and Research (to B.E.D.).

## Author contributions

All authors conceived of and designed the study and analysed and interpreted the results. F.A.B.v.M. carried out the experiments and wrote the manuscript, with substantial contributions to the writing from P.H. and B.E.D.

## Funding

## Competing interests

The authors declare no competing interests.

## Additional information

**Correspondence and requests for materials** should be addressed to Bas E. Dutilh.

# Reporting Summary

## Statistics

For all statistical analyses, confirm that the following items are present in the figure legend, table legend, main text, or Methods section.

| n/a | Confirmed | |
|---|---|---|
| ☐ | ☒ | The exact sample size (*n*) for each experimental group/condition, given as a discrete number and unit of measurement |
| ☐ | ☒ | A statement on whether measurements were taken from distinct samples or whether the same sample was measured repeatedly |
| ☐ | ☒ | The statistical test(s) used AND whether they are one- or two-sided<br>*Only common tests should be described solely by name; describe more complex techniques in the Methods section.* |
| ☐ | ☒ | A description of all covariates tested |
| ☐ | ☒ | A description of any assumptions or corrections, such as tests of normality and adjustment for multiple comparisons |
| ☐ | ☒ | A full description of the statistical parameters including central tendency (e.g. means) or other basic estimates (e.g. regression coefficient) AND variation (e.g. standard deviation) or associated estimates of uncertainty (e.g. confidence intervals) |
| ☐ | ☒ | For null hypothesis testing, the test statistic (e.g. *F*, *t*, *r*) with confidence intervals, effect sizes, degrees of freedom and *P* value noted<br>*Give P values as exact values whenever suitable.* |
| ☐ | ☒ | For Bayesian analysis, information on the choice of priors and Markov chain Monte Carlo settings |
| ☐ | ☒ | For hierarchical and complex designs, identification of the appropriate level for tests and full reporting of outcomes |
| ☐ | ☒ | Estimates of effect sizes (e.g. Cohen's *d*, Pearson's *r*), indicating how they were calculated |

*Our web collection on statistics for biologists contains articles on many of the points above.*

## Software and code

Policy information about availability of computer code

| Data collection | The MGnify API (https://www.ebi.ac.uk/metagenomics/api/latest/) assessed via Python 3 using the jsonapi_client library (https://github.com/qvantel/jsonapi-client). |
|---|---|
| Data analysis | CheckM v1.0.7; iTOL; Python 3 using the following libraries: Numpy, Scipy, matplotlib, pandas, seaborn, scikit-learn v0.21.3, EMDUnifrac, scikit-bio v0.5.5, GSEApy v0.7.3, ete3 v3.1.1, Cartopy v0.17.0.<br>All custom code used for this manuscript is available on Zenodo at DOI: 10.5281/zenodo.7651594. A stand-alone script to calculate SNB is available on https://github.com/MGXlab/social_niche_breadth. |

For manuscripts utilizing custom algorithms or software that are central to the research but not yet described in published literature, software must be made available to editors and reviewers. We strongly encourage code deposition in a community repository (e.g. GitHub). See the Nature Portfolio guidelines for submitting code & software for further information.

## Data

Policy information about availability of data

All manuscripts must include a data availability statement. This statement should provide the following information, where applicable:

- Accession codes, unique identifiers, or web links for publicly available datasets
- A description of any restrictions on data availability
- For clinical datasets or third party data, please ensure that the statement adheres to our policy

All data analysed during this study are included in this article and its Supplementary Data Files, or available in public repositories. The selected samples from the MGnify resource (https://www.ebi.ac.uk/metagenomics/) are described in Supp. Data File 1. The cleaned taxonomic profiles based on this data are available in Supp. Data File 2. The selected genomes from the PATRIC database (https://www.bv-brc.org/) are described in Supp. Data file 5. Measures derived from the PATRIC genomes, and the EGGO (https://github.com/jlw-ecoevo/eggo) and TimeTree (https://timetree.org/) database are available in Supp. Data File 3.

## Human research participants

Policy information about studies involving human research participants and Sex and Gender in Research.

| | |
|---|---|
| Reporting on sex and gender | *Use the terms sex (biological attribute) and gender (shaped by social and cultural circumstances) carefully in order to avoid confusing both terms. Indicate if findings apply to only one sex or gender; describe whether sex and gender were considered in study design whether sex and/or gender was determined based on self-reporting or assigned and methods used. Provide in the source data disaggregated sex and gender data where this information has been collected, and consent has been obtained for sharing of individual-level data; provide overall numbers in this Reporting Summary. Please state if this information has not been collected. Report sex- and gender-based analyses where performed, justify reasons for lack of sex- and gender-based analysis.* |
| Population characteristics | *Describe the covariate-relevant population characteristics of the human research participants (e.g. age, genotypic information, past and current diagnosis and treatment categories). If you filled out the behavioural & social sciences study design questions and have nothing to add here, write "See above."* |
| Recruitment | *Describe how participants were recruited. Outline any potential self-selection bias or other biases that may be present and how these are likely to impact results.* |
| Ethics oversight | *Identify the organization(s) that approved the study protocol.* |

Note that full information on the approval of the study protocol must also be provided in the manuscript.

# Field-specific reporting

Please select the one below that is the best fit for your research. If you are not sure, read the appropriate sections before making your selection.

☐ Life sciences ☐ Behavioural & social sciences ☒ Ecological, evolutionary & environmental sciences

For a reference copy of the document with all sections, see nature.com/documents/nr-reporting-summary-flat.pdf

# Ecological, evolutionary & environmental sciences study design

All studies must disclose on these points even when the disclosure is negative.

| | |
|---|---|
| Study description | Generalists can survive in many environments whereas specialists are restricted to a single environment. Although a classical concept in ecology, niche breadth has remained challenging to quantify for microbes because it depends on an objective definition of the environmental conditions. Here, by defining the environment of a microbe as the community it resides in, we integrated information from over 22 thousand environmental sequencing samples to derive a quantitative measure of the niche, which we call 'social niche breadth'. At the level of genera, we explored niche range strategies throughout the prokaryotic tree of life. We found that social generalists include opportunists that stochastically dominate local communities, while social specialists are stable but low in abundance. Social generalists have a more diverse and open pan genome than social specialists, but we found no global correlation between social niche breadth and genome size. Instead, we observed two distinct evolutionary strategies, where specialists have relatively small genomes in habitats with low local diversity, but relatively large genomes in habitats with high local diversity. Together, our analysis shines data-driven light on microbial niche range strategies. |
| Research sample | 22,518 selected from the MGnify database. |
| Sampling strategy | A maximum of 1,000 random samples per biome annotated with pipeline version 4.1 of MGnify. |
| Data collection | MGnify API. |

| | |
|---|---|
| Timing and spatial scale | 2019-08-20 |
| Data exclusions | None |
| Reproducibility | We only used publicly available data. |
| Randomization | Not applicable, because we analysed the full set of environmental sequencing datasets together. |
| Blinding | Not applicable, because our data-driven approach was specifically designed to avoid interpreting data according to categorical annotations, which we show in the paper are both arbitrary and redundant. |

Did the study involve field work?  ☐ Yes  ☒ No

# Reporting for specific materials, systems and methods

We require information from authors about some types of materials, experimental systems and methods used in many studies. Here, indicate whether each material, system or method listed is relevant to your study. If you are not sure if a list item applies to your research, read the appropriate section before selecting a response.

## Materials & experimental systems

| n/a | Involved in the study |
|---|---|
| ☒ | ☐ Antibodies |
| ☒ | ☐ Eukaryotic cell lines |
| ☒ | ☐ Palaeontology and archaeology |
| ☒ | ☐ Animals and other organisms |
| ☒ | ☐ Clinical data |
| ☒ | ☐ Dual use research of concern |

## Methods

| n/a | Involved in the study |
|---|---|
| ☒ | ☐ ChIP-seq |
| ☒ | ☐ Flow cytometry |
| ☒ | ☐ MRI-based neuroimaging |

