## [Peer Review File · Nature Ecology & Evolution]

Peer Review Information

Journal: Nature Ecology & Evolution

Manuscript Title: A social niche breadth score reveals niche range strategies of generalists and specialists

Corresponding author name(s): F. A. Bastiaan von Meijenfeldt

Editorial Notes:

Reviewer Comments & Decisions:

Decision Letter, initial version:

29th September 2022

Dear Dr Dutilh,

Your manuscript entitled "On specialists and generalists: niche range strategies across the tree of life" has now been seen by four reviewers, whose comments are copied below. The reviewers have raised a number of concerns which we would like to see addressed in a revised manuscript before we can reach a final decision regarding publication in Nature Ecology & Evolution.

The reviewers' comments are all fairly straightforward, and most can be addressed with a substantial text revision and improved clarity of the figures. However, addressing the comments from Reviewer 4 (and to some extent Reviewer 1) should entail further analyses to convince the reviewers of the robustness of the approach to data gaps/limitations.

We therefore invite you to revise your manuscript taking into account all reviewer comments. Please highlight all changes in the manuscript text file with track changes or colour highlighting. Please also add line numbers for ease of reviewing.

- * Include a "Response to reviewers" document detailing, point-by-point, how you addressed each reviewer comment. If no action was taken to address a point, you must provide a compelling argument. This response will be sent back to the reviewers along with the revised manuscript.
- * If you have not done so already please begin to revise your manuscript so that it conforms to our Article format instructions at <http://www.nature.com/natecolevol/info/final-submission>. Refer also to any guidelines provided in this letter.
- * Include a revised version of any required reporting checklist. It will be available to referees (and, potentially, statisticians) to aid in their evaluation if the manuscript goes back for peer review. A

2revised checklist is essential for re-review of the paper.

[REDACTED]

Nature Ecology & Evolution is committed to improving transparency in authorship. As part of our efforts in this direction, we are now requesting that all authors identified as 'corresponding author' on published papers create and link their Open Researcher and Contributor Identifier (ORCID) with their account on the Manuscript Tracking System (MTS), prior to acceptance. ORCID helps the scientific community achieve unambiguous attribution of all scholarly contributions. You can create and link your ORCID from the home page of the MTS by clicking on 'Modify my Springer Nature account'. For more information please visit www.springernature.com/orcid.

[REDACTED]

Reviewer expertise:

Reviewer #1: Microbial genome sizes

Reviewer #2: Microbial niches

Reviewer #3: Microbial niches

Reviewer #4: Microbial community ecology

Reviewers' comments:

2Reviewer #1 (Remarks to the Author):

In this manuscript Bastiaan von Meijenfeldt et al. present the results of their work on using a large dataset of environmental sequencing data to explore niche range strategies in different microbial taxa. The approach they use is to compare the communities that each microbial taxa resides in and based on that define their niche breadth. This is an interesting approach and could potentially highlight the physicochemical dimensions of their niche or whether they have promiscuous or specific interactions with other members of the community. Having a score for social niche breadth sounds exciting but having a rigorous and consistent way to interpret this score is equally important. While this score could be a mean for relative comparisons, there are some inconsistencies in the provided discussions that need clarification.

In the results section there are remarks as low/high local diversity, low/high social niche breadth, low/high diversity specialists but the thresholds for these delineations are not clear to the reader. It is essential to clarify these in the text. I see the median value provided in Figure 2d but still don't know how the SNB values provided in the text should be interpreted. For example in this section of the results SNB values are very close and difficult to interpret without prior knowledge of ecological strategies of these taxa. "Taxa with relatively low SNB for their ranks include known specialists such as *Christensenella* (SNB = 0.24), but also the taxa *Pelagibacteraceae* (SNB = 0.25) and *Prochlorococcus* (SNB = 0.22), which hold some of the most abundant organisms on Earth. These taxa, known for their highly streamlined genomes, are found in aquatic samples with a uniform microbial composition (Supp. fig. 5b) and thus have a low SNB. The genus *Roseobacter* (SNB = 0.30), whose members are considered marine generalists with large genomes and a versatile metabolism, is found in more diverse samples (Supp. fig. 5b) and has an SNB close to the median of all genera." For example, different representatives of family *Pelagibacteraceae* dominate marine and freshwater ecosystems (SAR11 vs LD12 respectively) and the overall community of marine and freshwater ecosystems are rather different. One would argue that at the family level *Pelagibacteraceae* should be rather generalist than specialist.

The title of this section "Generalists compete for dominance, specialists are stable at low abundance" does not seem to corroborate with the dominance of what you identify as specialists in the paragraph before (i.e., *Pelagibacteraceae* and *Prochlorococcus*)

I highly recommend authors to add a brief description of the type of data they obtain from MGnify regarding the community composition and what are the main steps in these analyses and how it differs for metagenomes, amplicon datasets and metatranscriptomes. How these different analysis pipelines used for each as authors call "experiment type" will affect their comparative analysis. The alpha diversity obtained by metagenome and amplicon data for the same sample is not the same, many low abundance taxa might not be detected in the metagenome or multiple ribosomal operons could affect the abundance calculations based on amplicon analysis, etc... will such differences impact their analysis?

In the analyses provided in this manuscript each piece of data is collected from a different source and as I understand genomes are not reconstructed from the same datasets used for community comparisons how could this affect the interpretations of the pangenome, genome size, etc.

Methods:

I have some comments regarding the datasets used in this study.

3-In experiment type section it is confusing as to what authors mean by category "unknown"? this could be very confusing and could potentially affect the results in ways that are difficult to discuss or conclude.

-Also, I would not say that abundance calculations based on metagenomes and amplicon is comparable to those calculated from metatranscriptomes. In the metatranscriptome datasets, activity of each taxa is affecting the final representation of reads affiliated to this taxa whereas in the other two methods abundance is not affected by the activity per se. I would suggest removing metatranscriptomes from this study.

-What is the threshold for differentiating between saline and non-saline. Does this refer to marine salinity or hypersaline? Clumping a large range of salinity together would be misleading.

I understand all the limitations caused by unknown ranks and taxa in meta-analysis. It would be great if authors discuss how different approaches they have tested for dealing with the unknown taxa would affect their results and conclusions. This could be a section in the supplementary results and discussions.

Since in the submitted file there was no line number, I copy here the part of text I am referring to in each comment.

Minor comments:

-All figures are packed with datapoints but presented in very small sizes both for the main figures and the supplementary figures. I highly recommend authors to take advantage of the available space and make their plots larger and less compact.

-I find the first sentence of the abstract disconnected from the rest of the abstract text

-implying a low turnover rate and stable community composition of highly diverse habitats

Maybe this could be concluded if these datasets are timeseries of the same sampling point otherwise how the comparison of alpha and beta diversity could signal low turnover rate?

-Most annotated biomes have low beta diversity reflecting consistent microbial composition.

It could be reformulated to mention that these biomes have low beta diversity indexes in comparison with each other inside one biome

-We used the observation that microbiomes are biosensors

Please add references

-Niche breadth across the tree of life and at different taxonomic ranks

Prokaryotic tree of life

-The distribution of SNB differs per taxonomic rank, where high-ranking taxa tend to be generalists and low-ranking taxa specialists

I agree with the explanation authors provide after this sentence, but I still feel resistance towards how this sentence is formulated. Clumping these high taxonomic ranks specifically for ubiquitous phyla such as proteobacteria with high diversity of ecological strategies in their representatives could be very misleading. I recommend some reformulation in this paragraph.

-Indeed, the four best-studied phyla, Proteobacteria, Firmicutes, Bacteroidetes, and Actinobacteria...

I don't think the higher representation of these taxa is solely due to them being more studied. Most environments are dominated by representatives of these taxa in metagenomic studies as well.

- Candidate phyla have a low SNB compared to established phyla

What do you mean by established? Maybe validly described? Although I don't think having a candidate status has an actual effect on the SNB. Maybe it is a side effect of most these candidate phyla being

among rare taxa? I think here you mean candidate phyla radiation rather than candidate phyla in general?

- Phyla with relatively specialised genera include...

Please provide stats on how prevalent specialized genera are in these phyla compared to the phyla with more generalized genera

-In conclusion, species in specialist genera are genomically more similar than species in generalist genera

What do you mean here, genome size? Pangenome size? Genomic content and functions?

-in the section about coding density and streamlining please add the name of relevant taxa. Are these specialists animal associated groups with streamlined genomes or free living groups with streamlined genomes such as SAR11? There are known differences in the streamlining route for these groups.

- as they require specific partners in their local communities

I am curious as your data could provide any hint towards whether these groups have specific hosts or rather a diverse range of partners in different environments?

-In cases where identical genomes had different taxonomic taxa

What do you mean by taxonomic taxa? Taxonomic affiliation maybe?

-A function was considered present in a species if at least 50% of its genomes contained it.

This sounds odd to me. Do you mean as a part of the core or in general? This threshold needs more explanation to make the logic clearer.

-carried out via the false discover rate (FDR) as suggested in discovery

Reviewer #2 (Remarks to the Author):

Comments to the Author

The article entitled "On specialists and generalists: niche range strategies across the tree of life" reports an analysis of environmental sequencing datasets publicly available to assess for each possible taxon the degree of variation of its surrounding community. To do so, the authors introduce a new metrics named SNB for Social Niche Breadth reflecting the community similarity between samples in which a specific microbial taxon occurs, with this score being low for organisms always detected in compositionally similar samples (called here social specialists) and this score being the highest for organisms that are found in sample with distinct community structure (social generalist). The SNB scores are then analysed in regards to biomes, metabolism, genomic features and compared to previous work on generalist and specialist features and (a little bit) to the ecological theory.

The implementation of this new metric is exciting and opportune in the meta'omic era, and could be more easily adopted by the research community with some improvements of the manuscript, that are developed in the following, and that could also include a conceptual schema of SNB.

The objectives of this work could also be more clearly stated, to clarify if the aim is here introducing a new metric and comparing it to the ones currently in use, or if it is to extract new features for understanding generalist/specialist (evolutionary strategy, ...). The postulates related to the creation of SNB score could also be formalised.

The article is well concise and globally easy to read, with the notable exception of the partitioning of

5the main text and supplementary text that has to be revised in my opinion. The English is fluid and no typos were noted.

Major comments

1 - While 'Social niche breadth (SNB)' is often stated in the text, 'Niche breadth' is also used throughout the manuscript, with sometimes an ambiguity between the "metabolic" niche breadth (the historical definition of niche breadth, i.e. similarly to Hutchinson niche; also sometimes called "general niche breadth" in the manuscript) and the SNB introduced in this work. Similarly, a careful revision of the text where, 'generalist' and 'specialist' are renamed 'social generalist' or 'social specialist' when opportune (similarly to 'habitat generalist' or 'habitat specialist' in other publications to differentiate with the historical names of generalist/specialist that refers to observations of the metabolic flexibility) would clarify the manuscript. (e.g. "for the vast majority of taxa, the SNB score is lower than expected based on random permutations (Fig. 2a-c), indicating that all microbes are *[SOCIAL]* specialists to some extent because they occur in a non-random subset of all samples."; "Whereas local communities are typically dominated by generalists, the relative abundance of generalists is more variable across samples than that of specialists, whose abundance is relatively stable.": Are you here mentioning *social* generalists and specialists accordingly to some of your results, or is it a general statement, that would then need a reference?). Similarly, 'diversity', 'alpha-diversity' and sometimes 'richness' are used. The definition of 'diversity' is so much dependent of the authors (and the readers!) that I would advise to uniquely use one of the latter.

2 - Some concepts or justifications are not well enough developed (or referenced), and data interpretations are not always in the main text. Mainly, expressing clearly the rationales and hypothesis behind the new concept of SNB would be necessary to fully exploit the provided discussion and comparison to previous study (e.g. relationship between niche breadth and genome size, ...)

a- Information on SNB score are all at the end of the manuscript in the methods section. Although it is the right place to present the formula, a bit of background (for example stating that SNB is a dissimilarity score based on Spearman correlation) would be useful at the beginning of the results, where the score is introduced. Indeed, most of the recent publications treat social interactions using networks and metrics based on graph theory, and it would be useful making clear very early on that this is not the case here.

b- How is SNB score filling a gap in ecological studies? How does it compare to trophic niche breadth or habitat niche breadth, etc, that are named in the introduction? How does the SNB score compare or differ from co-occurrence networks or any other microbial interactions network used in a massive number of recent studies? Why is SNB read out more accurate / appropriate to target ecological questions such as deterministic vs neutral assembly processes mentioned in the introduction? Etc. Please, describe better the advantages of SNB in the context of the state of the art.

c- The traditional definition of ecological niche breadth includes the fundamental and the realized niche. This latter is the n-dimensional space that a taxon actually inhabits in the presence of interspecific competition. Similarly, are you scoring the fundamental social niche breadth? I have the sensation that you are hypothesizing that the sum of all realized social interactions found in your 22 thousand of sample results in the fundamental social niche? Have you checked if the number of samples allows to 'saturate' the list of social partners for at least most of the taxon?

6- d- What are the postulates in the background of SNB? Probably at least that genetic information of the SSU summarize the metabolic capacity of an organism and that metabolic capacity determines the biogeographical distribution of the taxon. Also, a taxon is considered actively inhabiting in a biome sample if DNA is in sufficient quantity to produce at least 1 read per 10 000. Etc
- e- Strong statement such as "Also, it remains difficult to quantify the degree of similarity between categorical biomes." would deserve a justification or a good reference.
- f- Data interpretations better read in the text than in the figure captions.

3 - Data analysis:

- a- The paragraph "Taxa with relatively low SNB for their ranks include known specialists such as *Christensenella* [...] also have a high SNB, highlighting the diversity of their anaerobic habitats." is comparing SNB score-based assessment of niche breadth with more traditional metabolically-based niche breadth assessment. This could make a lot more sense if the hypothesis behind SNB and how it compared to previously introduced assessment of lifestyle would be discussed first (see also comment 2b and 2c). Is this paragraph, for the authors, a kind a validation that their SNB score is accurate to determine niche breadth?
- b- "... indicating that genomic signatures of SNB are generalizable across habitat and phylogeny": This hold true only if you demonstrate first that SNB scores have a genomic signature.
- c- "...at the genus rank we found that specialists are younger than generalists (Fig. 3b)": Can you explain how you conclude that from this figure? I was unable to track this information.
- d- "The correspondence between community heterogeneity and genomic heterogeneity confirms ...": I don't know where to find that in the data. What are the metrics used to assess "community heterogeneity" (SNB?) and "genomic heterogeneity" (heterogeneity of the independent genomic features?). Which statistical test was used?
- e- The final result section "Generalist pan genomes reflect fluctuating habitats, low and high diversity specialists differ in metabolic adaptations", social niche breadth is compared to metabolic niche breadth. This is a very important step to know if SNB is a metrics that allow to characterize the specificity of community assembly or of SNB can also be used as a proxy for metabolic niche breadth. However, no clear conclusion are made. The related figure 5 is also difficult to examine: would it be possible to have the data in a bar chart, with bar length being the 'fraction of genome' and the orientation of the bar indicating if enriched in social specialist (negative) or social generalist (positive)?
- f- Maybe more of curiosity questions: have you seen a difference in average SNB at the domain level between archaea and bacteria, as archaea are supposedly a very specialized domain? It seems that you have sample with no bacteria (22 518 – 22 295 = 223), are they archaea-only samples?

4-Figures, Figure legends, Figure captions and figure citations can be slightly improved for a better readability of the work, in particular Figure 5 (please see comment 3e).

- a- To shorten the rather long legends of this manuscript, I would recommend to only keep the descriptions of each panel and remove all interpretations from the legends. For example, in the legend of figure 1 (but it apply to all other figure legends), all the following cited sentence are data interpretation and not figure description and, in my point of view, would better fit in the main text: "Samples from similar annotated biomes cluster together based on..."; "Samples are separated by host-association and salinity"; "Most samples from the plants biome are derived" from seagrasses and macro algae from kelp forests, which is why they cluster near marine samples"; "Taxa richness differs per annotated biome and taxonomic rank. The low number of annotated species is a consequence of a

7relatively unexplored biosphere". "Annotated biomes with high alpha diversity have low beta diversity."

b- In figure 2b, and related text, an upper taxonomical level is always more generalist than its daughter taxon. This is very much expected as mentioned in the text. However, it leads to a representation where it is almost impossible to compare SNB scores across a taxonomical level due to the color gradient constrain (phyla are all orange, species are all blue). Once the fact that "high-ranking taxa tend to be generalists and low-ranking taxa specialists", wouldn't it be useful to somehow re-scale the color to use the full color gradient to each level to be able to visualize if, towards the leaves, some branches of the tree are more made of social generalist, social specialist or a mix of both?

c- As mentioned in comment 3e, the figure 5 is difficult to read. Another representation of the data might be envisaged.

5 – Methods:

a- The partitioning of information in the Methods section in the main text and in supplemental should be revised. For example, 10 methods to measures community composition dissimilarity are presented in the main text, but only one is used. It is not said what are the 9 others for. The 9 others methods should rather be presented somewhere else, and instead explain in the main text that a comparison of different metrics was done and is presented in the supplementary material, and that Spearman was selected as it performs well.

Another example is the presentation of 3 approaches to deal with the missing taxonomic ranks, presented in the main text, but only methods (ii) is used in the following. Similarly, method (ii) can be described in the main text while mentioning that other methods were tested and pointing to the supporting information.

b- The rather well accepted sparCC methods (<https://doi.org/10.1371/journal.pcbi.1002687>) has been demonstrated to be very robust to compositional data, and sometimes better than bray-curtis, spearson or pearson. On which criteria were the 10 presented methods selected?

6 - No discussion is provided about the fact that a DNA sequence is not the demonstration of the active presence of an organism. Yet, the active presence is included in the original definition of ecological niche and is partially used by the authors "the range of conditions in which an organism can live". (In soil for example, as much as 50-80% of organic carbon derived from microbial necromass, which include a large fraction of DNA). This issue can probably not be solved easily here, but should at least be mentioned. Are there major differences between samples for which metatranscriptome is available?

Minor comments

1- "tree of life" in the title and elsewhere make readers expect something different than what is presented in this work. Please rephrase to avoid this kind of overselling.

2- Remove the terms such as "as expected", "strikingly", etc.

Introduction:

3- Are microbiomes really capable of detecting disease state? Isn't it rather the disease state and its biochemical consequences that select for a different microbiome? Similarly, are microbes biosensors or

biomarkers?

4- "organisms that occur in diverse sample": Does 'diverse' mean here with a high biodiversity/richness or in a high number of sample or even compositionally different?

Results

5- "..., implying a low turnover rate...": Do you mean a turn over in the species composition (invasion, extinction, ...) or at the cell level (generational turn over)?

6- "Most annotated biomes have low beta diversity reflecting consistent microbial composition": I am a bit uncomfortable with this statement, as 'consistent' is a bit too strong. It can also be that number of taxon of the accessory community is small relative to the number of taxon making the core community, but this can represent a large number of taxon in absolute number.

7- "We used the observation that microbiomes are biosensors ...": observation from who? Depending, please give a reference.

8- "Only recently discovered, candidate phyla generally require specific growth conditions and are thus difficult to cultivate, consistent with their low SNB." The link between being difficult to cultivate and low SNB is unclear to me as the authors as using sequencing data from environmental sample, and as such, that did not require cultivation steps.

9- "have relatively generalised genera": please rephrased.

10- "At the generalist end of the spectrum are ubiquitous taxa": Do you mean cosmopolitan? Is there a reference for that?

11- "...while specialists are adapted to become dominant within their habitats.": Please, add 'in stable conditions' or something similar.

12- "Whereas local communities are typically dominated by generalists": what is a local community in the context of your work?

13- "Specialists have a low but constant abundance near carrying capacity (K-selected), and some (but not all) generalists are opportunistic taxa that reach high relative abundance when circumstance permits (r-selected)": can you please add a reference, maybe specific to prokaryotes?

14- At first instance "genomic features" is used in the text, please add a few words to explain what is meant, and how many independent features are considered.

15- "...but at the genus rank we found that specialists are younger than generalists (Fig. 3b)". Figure 3b is displaying the data at the family level, so at the taxonomical rank just above. So is it at the family of genus rank that social specialists are found younger than social generalists?

16- "because genome size estimates are often based on cultivated microbes that differ markedly from environmentally derived genomes, previous suggestions of a positive correlation between genome size and niche range are likely biased": please rephrase this sentence, as every work has its own inherent bias, including the present work.

17- "...for example, the genus Polyangium with a mean genome size of 12.7Mb." Can you add the SNB score too, so it reads "for example the genus Polyangium with a mean genome size of 12.7Mb and a SNB of XX".

Methods

18- "Taxa that were represented by less than 5 reads were removed before dissimilarity calculations": 5 reads in the total dataset or per sample?

19- "For some of the ecological dissimilarity": which ones?

20- "pairwise comparisons can be based on few taxa": Do you mean that it is the case "pairwise

9comparisons was based on a reduced number taxa", or that even with a reduced number of taxa the analysis is robust or meaningful "pairwise comparisons can be computed albeit based on few taxa"?

Displays

21- "Both colour-coding and size of markers represents the Shannon entropy": Why depicting the same metrics twice?

22- "Blue lines indicate samples with low alpha diversity and yellow lines indicate samples with high alpha diversity": Would it be possible to add a color scale as the alpha-diversity is here represented by a color gradient.

23- "Annotated biomes are arranged according to alpha diversity": is it the cumulative alpha-diversity of all samples of the biome (Boolean 'or') or the average alpha diversity?

Reviewer #3 (Remarks to the Author):

The authors present a very interesting and original study that uses the distribution of taxa with respect to overall microbiome composition in diverse natural habitats on a large scale as a proxy for their ecological niche breadth. They take advantage of the huge amount of existing data to bring some standard approaches (e.g. the study of co-occurrence of different species/taxa) to an entirely new scale, both data-wise and conceptually. This results, I believe, in an important empirical contribution. It uncovers many new patterns, some of them expected (larger pan-genome in generalists than in specialists) and other, largely unexpected and surprising given common assumptions in the field, such as the dominance of generalists in terms of abundance. Overall, the methodology is state of the art, and its robustness is shown using appropriate controls, random permutation tests, and comparison to alternative technical solutions.

The study is largely descriptive, and I tend to favor more hypothesis-driven analyses. But I do believe that in this case, the novelty of the observed patterns has an important value by itself, and studying the principles and mechanisms that produce them would require substantial work that is probably beyond the scope of this study. For instance, the authors discuss the differences in genomic content and size between specialists and generalists in low vs high diversity communities, as well as functional enrichment patterns, to a somewhat coarse-grained extent, which is probably a limitation imposed by the large scale of the study.

In summary, I enjoyed reading the paper. I think it is an important contribution to the field and opens many new questions and opportunities for future work. If any criticism, I found some figures very data-dense, which somewhat hinders their straightforward visual interpretation (e.g. fig 4d or Fig 5). Especially in a study of this scale, I like figures that summarize the main, big-picture findings in a more graphically interpretable way (with more detailed figures and of course the data itself available as a supplement to those that want to delve into the details). But this is a matter of personal choice, and of course, it is up to the authors how they present their data.

Djordje Bajic

10Reviewer #4 (Remarks to the Author):

The manuscript by von Meijenfeldt and colleagues proposes a community diversity-based metric for identifying ecological generalists and specialists. The main idea is that species living in very similar communities are likely to be specialises while those living in communities with diverse compositions are generalists. The underlying assumption is that the community diversity is an indicator of the environmental selection pressures. Once this definition is accepted, the authors use it to investigate distinguishing features of generalists compared to those of the specialists. This revealed differences in abundance profiles (generalists "stochastically" dominating while specialists being at low but constant abundance) as well as in their pangenome structures.

Overall, we find the topic highly interesting and of potential broad interest. The conclusions, however, require much stronger support.

Main points:

1. The central pillar of the study is the metric termed 'social niche breadth' (SNB). Yet, the manuscript is quite shallow on the justification of the used formula and its potential pitfalls. A rank correlation based metric needs to be used with much caution when several members of the community are very low abundance (a typical case in many habitats including soil). In such cases, the technical artifacts (DNA extraction, amplification, data processing etc.) mean that the estimated abundances are only order of magnitude estimates with ranks having no real meaning. How sensitive are the results to use if this metric? Using a combination of different dissimilarity metrics will be much more robust.
2. Another potential confounder is the niche alteration by microbes – many microbes secrete small molecules like amino acids that can alter the niches. The implications of such metabolic interactions on the analysis should be discussed as it can underestimate the true habitat diversity.
3. The manuscript seems to question the "classic" ecological theory that specialists have fitness advantage at the expense of versatility. Yet, this should be more prominently discussed. What explanations the data/analysis offers against the classical view? Both sides of the arguments should be clearly laid out and potential pitfalls discussed. An orthogonal analysis of longitudinal datasets will be important in this context.
4. The choice of including just a few meta-transcriptomics datasets is puzzling, it does not add anything to the study but rather adds potential confounder. Further, it is unclear how 16S vs shotgun metagenomics estimates are pooled together. I recommend analysing the two separately to check the robustness of the conclusions.
5. The manuscript is poor on clearly reporting the effect sizes, which are modest in many cases and thus potentially sensitive to the choices of the metrics and any biases in the data.
6. Repetition of the main analyses in a subset of habitats would be very insightful in testing the robustness/generalisation of the conclusions drawn.
7. Given the pan-genome level differences between generalists and specialists, it becomes imperative to discuss whether the "true functional taxonomic unit" should be the sub-species/strain rather than species or higher.
8. The enrichment analyses in the last figure are way to descriptive without any insights. I suggest

11moving the long lists to supplementary while presenting a noteworthy summary in the main figure/text.

9. Introduction would benefit from explicit mentioning of classic ecological theory re. niche occupation & competitive exclusion (terms of these theories are referred to in the Results section, such as r-strategists and K-strategists, but would be good to briefly address in introduction). A more substantial explanation of what falls under the term 'niche breadth' and the literature usage of this will also be important to include.

10. "To test this hypothesis, we compared SNB of microbes to their predicted maximal growth rates": the growth rate prediction is a tricky business. Why not use growth rates calculated from metagenomic samples with tools like coptr, iRep etc.?

11. Terms like "low diversity specialists" are unclear and warrant clear definitions.

12. Figure captions are sometimes unclear (e.g. Fig.2 panel e: what is the key for the letters?).

Sometimes citations to the supplementary are spelled out completely or shortened. Consistency might be good.

13. Regarding the genomes in generalist genera being more variable in size, what is the hypothesis? What is the literature suggesting? What functions are coded on these variable regions?

14. Please consider making a github repository to share the code and the results

15. "Lachnospiraceae (SNB = 0.50) and its genera (Supp. table 3), obligate anaerobes that were previously regarded as specialists 23 also have a high SNB" :Why Lachnospiraceae have a high SNB, where the diversity of their habitats come from? In general these concepts of "high diversity habitats" and "low diversity habitats" are not completely clear.

16. "(but see 48)" Please state explicitly what authors what to point out fro the reference.

*****END*****

Author Rebuttal to Initial comments

Response: We thank the reviewers for their helpful comments as we believe they have made our manuscript clearer and the conclusions more robust. Our revised manuscript file contains a substantially revised Main Text and Supplementary Results and Discussion. We have used "track changes". The revised version includes a number of new analyses that we have performed in response to the reviewers' suggestions. In addition to improving our existing figures, we have added 3 new figures, 4 new tables, and 13 new references.

Reviewers' comments:

Reviewer #1 (Remarks to the Author):

In this manuscript Bastiaan von Meijenfeldt et al. present the results of their work on using a large dataset of environmental sequencing data to explore niche range strategies in different microbial taxa. The approach they use is to compare the communities that each microbial taxa resides in and based on that define their niche breadth. This is an interesting approach and could potentially highlight the physicochemical dimensions of their niche or whether they have promiscuous or specific interactions with other members of the community. Having a score for social niche breadth sounds exciting but having a rigorous and consistent way to interpret this score is equally important. While this score could be a mean for relative comparisons, there are some inconsistencies in the provided discussions that need clarification.

In the results section there are remarks as low/high local diversity, low/high social niche breadth, low/high diversity specialists but the thresholds for these delineations are not clear to the reader. It is essential to clarify these in the text.

Response: We have clarified in the text what we mean with high and low. In some cases, high and low are empirical, i.e. based on correlative trends and therefore represent a relative qualification rather than an absolute threshold. We have clarified this in the text as appropriate. For example, we changed: "generalist genera often outcompete specialists in their local communities" to "SNB positively correlates with abundance within samples, meaning that generalists locally outcompete their more specialist neighbours".

I see the median value provided in Figure 2d but still don't know how the SNB values provided in the text should be interpreted. For example in this section of the results SNB values are very close and difficult to interpret without prior knowledge of ecological strategies of these taxa. "Taxa with relatively low SNB for their ranks include known specialists such as Christensenella (SNB = 0.24), but also the taxa Pelagibacteraceae (SNB = 0.25) and Prochlorococcus (SNB = 0.22), which hold some of the most abundant organisms on Earth. These taxa, known for their highly streamlined genomes, are found in aquatic samples with a uniform microbial composition (Supp. fig. 5b) and thus have a low SNB. The genus Roseobacter (SNB = 0.30), whose members are considered marine generalists with large genomes and a versatile metabolism, is found in more diverse samples (Supp. fig. 5b) and has an SNB close to the median of all genera."

For example, different representatives of family Pelagibacteraceae dominate marine and freshwater ecosystems (SAR11 vs LD12 respectively) and the overall community of marine and freshwater ecosystems are rather different. One would argue that at the family level Pelagibacteraceae should be rather generalist than specialist.

Response: We agree that it is difficult to interpret the scores at different ranks, because of the way the SNB is defined as the mean inverse Spearman correlation of the communities where the taxon is found, and higher-ranking taxa of course occur in more communities than lower-ranking taxa. For this reason, we have substantially rewritten this section and included a new measure (the modified Z-score) to quantify and compare SNB scores at each rank, where a negative Z-score indicates that a taxon has a low SNB for its rank and a positive Z-score that its SNB is relatively high. This is now explained in lines 125-128.

We have also added a section about Pelagibacteraceae: "While the family Pelagibacteraceae contains both marine and fresh-water representatives (in the SAR11 and LD12 clades, respectively {10.1093/molbev/msp239}), in our dataset it is found primarily in marine samples (Supp. fig. 5b). This highlights that future sampling of even more habitats, combined with more sensitive detection methods could change or refine SNB scores for some taxa." In multiple places in the text, we explain about Prochlorococcus as example cases.

The title of this section "Generalists compete for dominance, specialists are stable at low abundance" does not seem to corroborate with the dominance of what you identify as specialists in the paragraph before (i.e., Pelagibacteraceae and Prochlorococcus)

Response: We agree, and have added a sentence to clarify that our results are general correlations that contain exceptions, with Prochlorococcus as an example: "While these are general results based on correlations across samples, an exception is Prochlorococcus, which has a low SNB but a high local abundance (mean relative abundance 0.63%). This genus is among the top 10% of most abundant genera (Supp. table 3) and in the majority of its samples belongs to the top 20% of genera in terms of local abundance."

I highly recommend authors to add a brief description of the type of data they obtain from MGnify regarding the community composition and what are the main steps in these analyses and how it differs for metagenomes, amplicon datasets and metatranscriptomes. How these different analysis pipelines used for each as authors call "experiment type" will affect their comparative analysis. The alpha diversity obtained by metagenome and amplicon data for the same sample is not the same, many low abundance taxa might not be detected in the metagenome or multiple ribosomal operons could affect the abundance calculations based on amplicon analysis, etc... will such differences impact their analysis?

Response: We have substantially rewritten the Supplementary Results and Discussion with an introduction on the different experiment types, and have added a new analysis ('Robustness to experiment type') where we verify our general results based on four subsets of the data, from each of the four different experiment types. This analysis showed that even though the results based on samples from a single experiment type sometimes differ for specific taxa, our overall results are robust. The observed differences are mostly due to differences in the sampled environments, rather than differences in the experiment type, highlighting the value in using samples from many different environments, as we did.

In the analyses provided in this manuscript each piece of data is collected from a different source and as I understand genomes are not reconstructed from the same datasets used for community comparisons how could this affect the interpretations of the pangenome, genome size, etc.

Response: the reviewer is right that the genomes in the PATRIC database are from a different source than the MGnify taxonomic annotation pipeline, and sometimes genus rank genomic inferences are based on strains that are not found in MGnify. While we have no easy way of testing whether the strains are exactly the same (they are most likely not, as most organisms in MGnify do not have a species annotation), we assume that the genomic content of the organisms in both databases is at least somewhat comparable at the genus rank. We have made it clearer in the text that these data sources are different, by explicitly naming the databases (PATRIC, EGGO, and TimeTree) and stating that comparisons are made at the genus rank. In addition, we now mention how many genera are shared between MGnify and PATRIC: "The PATRIC database contains genome sequences for 1,704 out of the 2,133 genera that we investigated in our global microbiome dataset (Supp. table 3). Although these genomes likely belong to different strains or species than those observed in MGnify, we reduce the inconsistencies in our analysis by assessing their genomic features at the genus rank."

Methods:

I have some comments regarding the datasets used in this study.

-In experiment type section it is confusing as to what authors mean by category "unknown"? this could be very confusing and could potentially affect the results in ways that are difficult to discuss or conclude.

Response: We agree and have added a section to the Supplementary Results and Discussion about the different experiment types (see previous comments) and why we believe they can all be included (i.e. since they are all annotated with the same taxonomic pipeline). The newly added analysis where we compare the different experiment types also includes the 'unknown' type and shows the effect of only including these samples on our final results.

-Also, I would not say that abundance calculations based on metagenomes and amplicon is comparable to those calculated from metatranscriptomes. In the metatranscriptome datasets, activity of each taxa is affecting the final representation of reads affiliated to this taxa whereas in the other two methods abundance is not affected by the activity per se. I would suggest removing metatranscriptomes from this study.

Response: Our newly added analysis where we compare the different experiment types shows that most results are qualitatively similar when using only metatranscriptomic samples, compared to including other experiment types. Where they are not, inferences are difficult to make because of the very small sample size. For example, metatranscriptomic samples with low alpha diversity do not show a positive correlation between SNB and genome size, however, these are only 4 samples and they fall within the level of noise that we see for other experiment types. We agree that theoretically, removing them from the start might have been a good idea, but these results do not convince us that this is necessary at this stage, which would entail a lot of work in remaking datasets, analyses, figures, and text. Since there are only 13 metatranscriptomic samples (<0.06%), removing them would only very slightly change our final results.

-What is the threshold for differentiating between saline and non-saline. Does this refer to marine salinity or hypersaline? Clumping a large range of salinity together would be misleading.

Response: We observe that the samples cluster according to salinity based on the t-SNE and PCOA projections, but we do not use this differentiation anywhere except in colour-coding of figures that

include the 'environment type'. This assignment is based on the biome annotations, see Supp. fig. 3 for details.

I understand all the limitations caused by unknown ranks and taxa in meta-analysis. It would be great if authors discuss how different approaches they have tested for dealing with the unknown taxa would affect their results and conclusions. This could be a section in the supplementary results and discussions.

Response: We have added a section to the Supplementary Results and Discussion about the approaches for dealing with unknowns, which we partly moved from the Methods section and have also extended. In addition, we now more clearly mention that different ways of dealing with the unknowns give nicely correlating results (Supp. fig. 6e).

Since in the submitted file there was no line number, I copy here the part of text I am referring to in each comment.

Response: We forgot to add line numbers in the first submission but have now done that.

Minor comments:

-All figures are packed with datapoints but presented in very small sizes both for the main figures and the supplementary figures. I highly recommend authors to take advantage of the available space and make their plots larger and less compact.

Response: We have changed the main figures wherever possible and have increased their size by ~25%. We also want to stress that the final figures will be vector images instead of pixels so zooming in will be possible.

-I find the first sentence of the abstract disconnected from the rest of the abstract text

Response: The first sentence is to explain in general terms what are "generalists and specialists", the two most important words in the Title, and ease the reader into the concept of "niche breadth". We agree that it could be phrased better and have changed it.

-implying a low turnover rate and stable community composition of highly diverse habitats
Maybe this could be concluded if these datasets are timeseries of the same sampling point otherwise how the comparison of alpha and beta diversity could signal low turnover rate?

Response: We have removed the mentioning of turnover rates from the manuscript because it caused confusion.

-Most annotated biomes have low beta diversity reflecting consistent microbial composition.
It could be reformulated to mention that these biomes have low beta diversity indexes in comparison with each other inside one biome

Response: We have rewritten to: "Most samples from the same annotated biome are relatively similar as reflected by a low beta diversity."

-We used the observation that microbiomes are biosensors
Please add references

Response: We have added five references.

-Niche breadth across the tree of life and at different taxonomic ranks
Prokaryotic tree of life

Response: We have changed "tree of life" to "prokaryotic tree of life" wherever feasible, while keeping in mind the readability of the text.

-The distribution of SNB differs per taxonomic rank, where high-ranking taxa tend to be generalists and low-ranking taxa specialists
I agree with the explanation authors provide after this sentence, but I still feel resistance towards how this sentence is formulated. Clumping these high taxonomic ranks specifically for ubiquitous phyla such as proteobacteria with high diversity of ecological strategies in their representatives could be very misleading. I recommend some reformulation in this paragraph.

Response: We very much agree with this suggestion. We have rewritten to "The distribution of SNB scores differs per taxonomic rank. High-ranking taxa tend to have higher SNB scores than low-ranking taxa (Fig. 2d,e), which intrinsically occur in a subset of the samples of their parent taxa." to prevent the formulation of calling high-ranking taxa generalists. We have also changed the next sentence to "High-ranking taxa either have high SNB scores ..." to get this point across.

-Indeed, the four best-studied phyla, Proteobacteria, Firmicutes, Bacteroidetes, and Actinobacteria...
I don't think the higher representation of these taxa is solely due to them being more studied. Most environments are dominated by representatives of these taxa in metagenomic studies as well.

Response: Indeed, this is correct; we also see this in our data. We have added that the four phyla "are dominant across a wide range of environments (Supp. fig. 17)".

- Candidate phyla have a low SNB compared to established phyla
What do you mean by established? Maybe validly described? Although I don't think having a candidate status has an actual effect on the SNB. Maybe it is a side effect of most these candidate phyla being among rare taxa? I think here you mean candidate phyla radiation rather than candidate phyla in general?

Response: We have rewritten this paragraph to clarify what we mean: "Phyla with the "candidatus" status have a low SNB compared to most validly described phyla (Supp. fig. 17)." and explain this in the following sentence: "The connection between the candidatus status and low SNB reflects a discovery bias of these phyla where widespread lineages tend to be discovered and described sooner than rare ones." We have also added that specific growth conditions "may be reflected in relatively stable specialised microbial communities" instead of "are difficult to cultivate".

- Phyla with relatively specialised genera include...

Please provide stats on how prevalent specialized genera are in these phyla compared to the phyla with more generalized genera

Response: We have now changed our description of these phyla to the modified Z-score which we think is easier to interpret.

-In conclusion, species in specialist genera are genomically more similar than species in generalist genera

What do you mean here, genome size? Pangenome size? Genomic content and functions?

Response: We have clarified in the text: "In conclusion, species in specialist genera are genomically more similar than species in generalist genera, reflected in more similar genome sizes and less variation in functions, and an associated smaller and more closed pan genome than generalist genera."

-in the section about coding density and streamlining please add the name of relevant taxa. Are these specialists animal associated groups with streamlined genomes or free living groups with streamlined genomes such as SAR11? There are know differences in the streamlining route for these groups.

Response: The specialists we are talking about are actually likely not streamlined (because generalists have a higher coding density). To make it clearer we have now reversed the specialists and generalists in this argument: "Coding density, a signature of genomic streamlining [10.1038/ismej.2014.60], is significantly lower in LD-specialists than in social generalists ($p < 0.003$, one-tailed T-test, measured in number of coding sequences per base pair with generalists defined as $SNB > 0.35$ (697 genera) and LD-specialists defined as present in samples with a mean alpha diversity ≤ 11 (552 genera))."

- as they require specific partners in their local communities

I am curious as your data could provide any hint towards whether these groups have specific hosts or rather a diverse range of partners in different environments?

Response: We agree that this is a very interesting question but feel that it was beyond the scope of this manuscript and this rebuttal. While in principle the data is there to address this, we note that the MGnify annotations often do not go down to the species rank, which might be important for the specific partner associations referred to by the Reviewer.

-In cases where identical genomes had different taxonomic taxa

What do you mean by taxonomic taxa? Taxonomic affiliation maybe?

Response: Yes, we meant taxonomic annotations; we have rewritten. We note that sometimes, a genome is present in PATRIC twice with two different species annotations. If this is the case, we discard the genome because we are uncertain of its taxonomic affiliation as mentioned in the Methods section.

-A function was considered present in a species if at least 50% of its genomes contained it.

This sounds odd to me. Do you mean as a part of the core or in general? This threshold needs more explanation to make the logic clearer.

Response: What we mean is that if a species has multiple genome strains in PATRIC, the function is only considered present if the majority of its genomes carried it. We have changed this to "if at least 50% of the genomes with this species annotation contained it in the PATRIC database." to make this clear.

-carried out via the false discover rate (FDR) as suggested in discovery

Response: Thanks :-).

Reviewer #2 (Remarks to the Author):

Comments to the Author

The article entitled "On specialists and generalists: niche range strategies across the tree of life" reports an analysis of environmental sequencing datasets publicly available to assess for each possible taxon the degree of variation of its surrounding community. To do so, the authors introduce a new metrics named SNB for Social Niche Breadth reflecting the community similarity between samples in which a specific microbial taxon occurs, with this score being low for organisms always detected in compositionally similar samples (called here social specialists) and this score being the highest for organisms that are found in sample with distinct community structure (social generalist). The SNB scores are then analysed in regards to biomes, metabolism, genomic features and compared to previous work on generalist and specialist features and (a little bit) to the ecological theory. The implementation of this new metric is exciting and opportune in the meta'omic era, and could be more easily adopted by the research community with some improvements of the manuscript, that are developed in the following, and that could also include a conceptual schema of SNB. The objectives of this work could also be more clearly stated, to clarify if the aim is here introducing a new metric and comparing it to the ones currently is use, or if it is to extract new features for understanding generalist/specialist (evolutionary strategy, ...). The postulates related to the creation of SNB score could also be formalised. The article is well concise and globally easy to read, with the notable exception of the partitioning of the main text and supplementary text that has to be revised in my opinion. The English is fluid and no typos were noted.

Major comments

1 - While 'Social niche breadth (SNB)' is often stated in the text, 'Niche breadth' is also used throughout the manuscript, with sometimes an ambiguity between the "metabolic" niche breadth (the historical definition of niche breadth, i.e. similarly to Hutchinson niche; also sometimes called "general niche breadth" in the manuscript) and the SNB introduced in this work. Similarly, a careful revision of the text where, 'generalist' and 'specialist' are rename 'social generalist' or 'social specialist' when opportune (similarly to 'habitat generalist' or 'habitat specialist' in other publications to differentiate with the historical names of generalist/specialist that refers to observations of the metabolic flexibility) would clarify the manuscript. (e.g. "for the vast majority of taxa, the SNB score is lower than expected based on random permutations (Fig. 2a-c), indicating that all microbes are

[SOCIAL] specialists to some extent because they occur in a non-random subset of all samples.”; “Whereas local communities are typically dominated by generalists, the relative abundance of generalists is more variable across samples than that of specialists, whose abundance is relatively stable.”: Are you here mentioning *social* generalists and specialists accordingly to some of your results, or is it a general statement, that would then need a reference?).

Response: We agree and have added ‘social’ as suggested to a lot of places in the text, while keeping the readability of the text in mind. To prevent very long wording we have changed our description of “high diversity and low diversity specialists” to “HD- and LD-specialists”.

Similarly, ‘diversity’, ‘alpha-diversity’ and sometimes ‘richness’ are used. The definition of ‘diversity’ is so much dependent of the authors (and the readers!) that I would advice to uniquely use one of the latter.

Response: We agree and have clarified whenever such a measure is mentioned, for example changed ‘alpha diversity of habitats’ to ‘mean alpha diversity of habitats’. See also Methods section where we describe the diversity measures and we mention that “When the terms alpha and beta diversity are used, we refer to first order diversity measures on the rank order unless otherwise stated.”

2 - Some concepts or justifications are not well enough developed (or referenced), and data interpretations are not always in the main text. Mainly, expressing clearly the rationales and hypothesis behind the new concept of SNB would be necessary to fully exploit the provided discussion and comparison to previous study (e.g. relationship between niche breadth and genome size, ...)

a- Information on SNB score are all at the end of the manuscript in the methods section. Although it is the right place to present the formula, a bit of background (for example stating that SNB is a dissimilarity score based of Spearman correlation) would be useful at the beginning of the results, where the score is introduced. Indeed, most of the recent publications treats social interactions using networks and metrics based on graph theory, and it would be useful making clear very early on that this is not the case here.

Response: We have extended the first paragraphs of the Results section to include an introduction to the SNB score:

“We assume that the small subunit rRNA (SSU) gene that is queried is a proxy for the genetic content of a taxon that defines its traits. Specific traits exist at all taxonomic ranks and determine their occurrence across microbiomes [10.1126/science.aac9323]. Since the taxonomic annotations are based on a reference taxonomy and the biosphere is underexplored, sometimes high-ranking taxa do not have low-ranking annotations like species (Fig. 1e). We considered that members of a taxon are alive and growing if they represented a relative abundance of at least 1/10,000 of the prokaryotic reads in a sample, and thus ignore the possibility of migration from other sources and the potential of dead organic matter contributing DNA to the sequencing results [10.1111/gcb.14781]. Next, we defined SNB as the mean pairwise dissimilarity between these microbiomes. After benchmarking 150 different ecological dissimilarity measures for their ability to separate the annotated biomes, we chose mean pairwise dissimilarity based on the inverse Spearman’s rank correlation of known taxa at taxonomic rank order to quantify SNB (Supp. results and discussion). Thus, taxa with a low SNB score are found in samples with very similar microbial composition (social specialists) and taxa with a high SNB score are found in dissimilar samples (social generalists). Our approach accounts for database biases, as some environments are much more frequently sampled than others (Fig. 1a, Supp. results

and discussion). Indeed, taxa that are detected in the same number of samples, or annotated biomes, may have very different SNB scores (Fig. 2a-c). Different from studies that investigate co-occurrence of taxa across samples {10.1371/journal.pcbi.1002606,10.1038/s41559-020-01353-4}, SNB quantifies the range of communities that a taxon can occur in. SNB treats each sample as a local niche, and infers that taxa that occur across highly differing communities are social generalist, while taxa that occur in similar communities are social specialists. Since SNB is calculated when a taxon is present over a detection limit of 1/10,000 reads, relative abundance and associated variability of a taxon's distribution are observables and can be associated to niche range, rather than part of the definition as in {10.1890/08-0851.1}.”

b- How is SNB score filling a gap in ecological studies? How does it compare to trophic niche breadth or habitat niche breadth, etc, that are named in the introduction? How does the SNB score compare or differ from co-occurrence networks or any other microbial interactions network used in a massive number recent studies? Why is SNB read out more accurate / appropriate to target ecological question such as deterministic vs neutral assembly process mentioned in the introduction? Etc. Please, describe better the advantages of SNB in the context of the state of the art.

Response: See also our response above: we have added a comparison to other niche breadth measures. Also, the SNB score shows very different results from simply counting presence-absence in different samples / environment types (Fig. 2a,b), which we believe is an important comparison.

c- The traditional definition of ecological niche breadth includes the fundamental and the realised niche. This latter is the n-dimensional space that a taxon actually inhabits in the presence of interspecific competition. Similarly, are you scoring the fundamental social niche breadth? I have the sensation that you are hypothesizing that the sum of all realized social interactions found in your 22 thousand of sample results in the fundamental social niche? Have you checked if the number of samples allows to 'saturate' the list of social partners for at least most of the taxon?

Response: We have added a section on the n-dimensional niche concept in the introduction to clarify how we envision the SNB score (as a proxy for the n-dimensional realised niche of a microbe that includes both environmental and intercommunity interactions):

“Niche breadth definitions that assess the full n-dimensional niche space {htj} have been based on occurrence in microbiomes. Rather than the theoretical fundamental niche, microbial occurrence represents its empirical realised niche. Because of complex interactions within microbial communities, the realised niche can both be smaller, e.g. due to competition {10.15698/mic2018.05.628}, or larger, e.g. due to metabolic dependencies {10.1038/s41559-021-01505-0}.”

And

“As such, we treat community composition as a proxy for the realised niche of a microbe, that reflects both the abiotic environment and the microbial interactions within. Similar reasoning has been used to quantify niche range of Eukaryotes without the use of external habitat definitions {10.1111/j.1365-2745.2007.01236.x}.”

d- What are the postulates in the background of SNB? Probably at least that genetic information of the SSU summarize the metabolic capacity of an organism and that metabolic capacity determines

the biogeographical distribution of the taxon. Also, a taxon is considered actively inhabiting in a biome sample if DNA is in sufficient quantity to produce at least 1 read per 10 000. Etc

Response: We have added these postulates to the first section of the Results where we describe the SNB score (see also our response above to suggestion 2a).

e- Strong statement such as “Also, it remains difficult to quantify the degree of similarity between categorical biomes.” would deserve a justification or a good reference.

Response: We have rewritten to: “Also, it is difficult to quantify the degree of similarity between categorical biomes in a biologically meaningful way.” As we believe this is inherently true. Mathematically categories are equidistant units, while biomes range in their similarity across many dimensions.

f- Data interpretations better read in the text than in the figure captions.

Response: We agree that any data interpretation in the figure caption should also be in the text and we have tried to clarify when needed. We believe that some data interpretation in figure captions is justified to make them better readable, especially given that some are arguably difficult to understand at face value.

3 - Data analysis:

a- The paragraph “Taxa with relatively low SNB for their ranks include known specialists such as *Christensenella* [...] also have a high SNB, highlighting the diversity of their anaerobic habitats.” is comparing SNB score-based assessment of niche breadth with more traditional metabolically-based niche breadth assessment. This could make a lot more sense if the hypothesis behind SNB and how it compared to previously introduced assessment of lifestyle would be discussed first (see also comment 2b and 2c). Is this paragraph, for the authors, a kind of validation that their SNB score is accurate to determine niche breadth?

Response: See our earlier comments. And yes, this paragraph is meant as a form of validation/justification/clarification of the SNB score based on specific examples.

b- “... indicating that genomic signatures of SNB are generalizable across habitat and phylogeny”: This hold true only if you demonstrate first that SNB scores have a genomic signature.

Response: We agree and have rewritten to: “indicating that genomic signatures of SNB (see below) are generalisable across habitat and phylogeny”.

c- “...at the genus rank we found that specialists are younger than generalists (Fig. 3b)”: Can you explain how you conclude that from this figure? I was unable to track this information.

Response: We have added “indicated by a consistent positive correlation between SNB and clade age”.

d- “The correspondence between community heterogeneity and genomic heterogeneity confirms ...”: I don’t know where to find that in the data. What are the metrics used to assess “community heterogeneity” (SNB?) and “genomic heterogeneity” (heterogeneity of the independent genomic features?). Which statistical test was used?

Response: We have rewritten to: "The correspondence between genomic heterogeneity and the heterogeneity of communities (SNB) confirms the strong association between ecological and genomic diversification."

e- The final result section "Generalist pan genomes reflect fluctuating habitats, low and high diversity specialists differ in metabolic adaptations", social niche breadth is compared to metabolic niche breadth. This is a very important step to know if SNB is a metrics that allow to characterize the specificity of community assembly or of SNB can also be used as a proxy for metabolic niche breadth. However, no clear conclusion are made. The related figure 5 is also difficult to examine: would it be possible to have the data in a bar chart, with bar length being the 'fraction of genome' and the orientation of the bar indicating if enriched in social specialist (negative) or social generalist (positive)?

Response: We agree that the previous figure was not easy to read, thanks for this good idea to improve it, we have changed Fig. 5 as suggested and rewritten the GSEA paragraph to clarify.

f- Maybe more of curiosity questions: have you seen a difference in average SNB at the domain level between archaea and bacteria, as archaea are supposedly a very specialized domain? It seems that you have sample with no bacteria (22 518 – 22 295 = 223), are they archaea-only samples?

Response: Yes, indeed some of our samples only contain archaea. These could be only archaeal environments or studies targeting specific archaeal gene regions and the metadata often does not allow as to assess this. This makes a comparison between the two different domains more tricky than for other taxa. We do observe that the vast majority of these archaeal samples are from marine settings:

Having said that, we think it's interesting that Euryarchaeota is one of the phyla with relatively generalised genera (median Z-score = 1.06).

4-Figures, Figure legends, Figure captions and figure citations can be slightly improved for a better readability of the work, in particular Figure 5 (please see comment 3e).

a- To shorten the rather long legends of this manuscript, I would recommend to only keep the descriptions of each panel and remove all interpretations from the legends. For example, in the legend of figure 1 (but it apply to all other figure legends), all the following cited sentence are data interpretation and not figure description and, in my point of view, would better fit in the main text: "Samples from similar annotated biomes cluster together based on..."; "Samples are separated by host-association and salinity"; "Most samples from the plants biome are derived" from seagrasses

and macro algae from kelp forests, which is why they cluster near marine samples”; “Taxa richness differs per annotated biome and taxonomic rank. The low number of annotated species is a consequence of a relatively unexplored biosphere”. “Annotated biomes with high alpha diversity have low beta diversity.”

Response: We have reduced some of the lengthy interpretations as suggested. We feel that some of the mentioned phrases are still justified as they contain important explanations on how the figure was created, or may be considered the title of the figure. We have rewritten the captions with this in mind.

b- In figure 2b, and related text, an upper taxonomical level is always more generalist than its daughter taxon. This is very much expected as mentioned in the text. However, it leads to a representation where it is almost impossible to compare SNB scores across a taxonomical level due to the color gradient constrain (phyla are all orange, species are all blue). Once the fact that “high-ranking taxa tend to be generalists and low-ranking taxa specialists”, wouldn’t it be useful to somehow re-scale the color to use the full color gradient to each level to be able to visualize if, towards the leaves, some branches of the tree are more made of social generalist, social specialist or a mix of both?

Response: We think this is a great suggestion and have changed a substantial part of the Results section based on this idea. We have made a new rank-specific score (the modified Z-score) to compare taxa at their own rank. To accommodate this change we have removed the previous Supp. Table 3 and replaced it with the modified Z-score table.

c- As mentioned in comment 3e, the figure 5 is difficult to read. Another representation of the data might be envisaged.

Response: See previous response, this is a nice idea and we have implemented accordingly.

5 – Methods:

a- The partitioning of information in the Methods section in the main text and in supplemental should be revised. For example, 10 methods to measures community composition dissimilarity are presented in the main text, but only one is used. It is not said what are the 9 others for. The 9 others methods should rather be presented somewhere else, and instead explain in the main text that a comparison of different metrics was done and is presented in the supplementary material, and that Spearman was selected as it performs well.

Response: We have substantially rewritten the Supplementary Results and Discussion to accommodate this. In addition, we discuss the different approaches for dealing with unknowns etc. In the Results section of the main text we have now included a paragraph introducing the SNB score itself and refer to the Supplement for further details (see earlier responses).

Another example is the presentation of 3 approaches to deal with the missing taxonomic ranks, presented in the main text, but only methods (ii) is used in the following. Similarly, method (ii) can be described in the main text while mentioning that other methods were tested and pointing to the supporting information.

Response: Please see our previous response.

b- The rather well accepted sparCC methods (<https://doi.org/10.1371/journal.pcbi.1002687>) has been demonstrated to be very robust to compositional data, and sometimes better than bray-curtis, spearson or pearson. On which criteria were the 10 presented methods selected?

Response: We agree that it would have been a good idea to include SparCC in the benchmark of diversity metrics. We selected the methods in the benchmark based on a reasonable overview of the diversity literature, but indeed it may be expected that some would still be missed. We would prefer not to add extra methods now, as it would be a lot of work to implement: we would need to calculate >250 million pairwise correlations between all datasets, before we even know if it has a (slightly) higher PERMANOVA F-score or not. We note that most of the distance metrics yielded similar results (see Supp. Fig. 6) so we do not expect to get qualitatively different results with sparCC.

6 - No discussion is provided about the fact that a DNA sequence is not the demonstration of the active presence of an organism. Yet, the active presence is included in the original definition of ecological niche and is partially used by the authors "the range of conditions in which an organism can live". (In soil for example, as much as 50-80% of organic carbon derived from microbial necromass, which include a large fraction of DNA). This issue can probably not be solved easily here, but should at least be mentioned. Are there major differences between samples for which metatranscriptome is available?

Response: This is a good point. We have now added these considerations to the Results section where we mention our postulates: "We considered that members of a taxon are alive and growing if the taxon represented a relative abundance of at least 1/10,000 of the prokaryotic reads in a sample, and thus ignore the possibility of migration from other sources and the potential of dead organic matter contributing DNA to the sequencing results {10.1111/gcb.14781}." In addition, we have now added the previously mentioned new analysis where we also include interpretation of our general results based on just the metatranscriptomic set of samples.

Minor comments

1- "tree of life" in the title and elsewhere make readers expect something different than what is presented in this work. Please rephrase to avoid this kind of overselling.

Response: We agree and have added 'prokaryotic' throughout the manuscript in many places, including the Abstract. We have kept the shorter phrase "tree of life" in the Title because we think it sounds much nicer, but we did change 'across' to 'throughout' the Tol, which should avoid overselling by making it clear that we do not sample the entire tree.

2- Remove the terms such as "as expected", "strikingly", etc.

Response: We have removed them all.

Introduction:

3- Are microbiomes really capable of detecting disease state? Isn't it rather the disease state and its biochemical consequences that select for a different microbiome? Similarly, are microbes biosensors or biomarkers?

Response: The reviewer is right that the microbiome is more a biomarker than a biosensor (but see references #31-35). We have changed it in the text.

4- "organisms that occur in diverse sample": Does 'diverse' mean here with a high biodiversity/richness or in a high number of sample or even compositionally different?

Response: We have changed to 'compositionally dissimilar'.

Results

5- "..., implying a low turnover rate...": Do you mean a turn over in the species composition (invasion, extinction, ...) or at the cell level (generational turn over)?

Response: We have removed 'turnover' across the entire manuscript because we agree it's a confusing term.

6- "Most annotated biomes have low beta diversity reflecting consistent microbial composition": I am a bit uncomfortable with this statement, as 'consistent' is a bit too strong. It can also be that number of taxon of the accessory community is small relative to the number of taxon making the core community, but this can represent a large number of taxon in absolute number.

Response: Yes, the reviewer is right that sometimes the absolute number of taxa that changes can still be very high in high alpha diversity/low beta diversity cases. We have rewritten to: "Notably, Annotated biomes with high mean alpha diversity such as soils had low beta diversity (Fig. 1f), implying a relatively stable core community across these high diversity habitats."

7- "We used the observation that microbiomes are biosensors ...": observation from who? Depending, please give a reference.

Response: We have added the references.

8- "Only recently discovered, candidate phyla generally require specific growth conditions and are thus difficult to cultivate, consistent with their low SNB." The link between being difficult to cultivate and low SNB is unclear to me as the authors as using sequencing data from environmental sample, and as such, that did not require cultivation steps.

Response: Yes, we have removed the cultivation link suggestion.

9- "have relatively generalised genera": please rephrased.

Response: We did.

10- "At the generalist end of the spectrum are ubiquitous taxa": Do you mean cosmopolitan? Is there a reference for that?

Response: We mean ubiquitous in our dataset and have rewritten accordingly: "At the generalist end of the spectrum are taxa that are ubiquitously present in our dataset (Fig. 2a,b) like the genera..."

11- "...while specialists are adapted to become dominant within their habitats.": Please, add 'in stable conditions' or something similar.

Response: We have added "in stable conditions".

12- "Whereas local communities are typically dominated by generalists": what is a local community in the context of your work?

Response: A sample, we rewrote.

13- "Specialists have a low but constant abundance near carrying capacity (K-selected), and some (but not all) generalists are opportunistic taxa that reach high relative abundance when circumstance permits (r-selected)": can you please add a reference, maybe specific to prokaryotes?

Response: We have added the original MacArthur and Wilson reference.

14- At first instance "genomic features" is used in the text, please add a few words to explain what is meant, and how many independent features are considered.

Response: We have rewritten to: "We set out to identify genomic features associated with social niche breadth using publicly accessible genome sequences from the PATRIC database {10.1093/nar/gkz943} (Supp. table 5). These features include the mean genome size of all species in the genus, the variation in these genome sizes, the pan genome size (i.e. the total number of functions present in all genomes), and the pan genome openness (calculated as the pan genome size divided by the mean genome size)."

15- "...but at the genus rank we found that specialists are younger than generalists (Fig. 3b)". Figure 3b is displaying the data at the family level, so at the taxonomical rank just above. So is it at the family of genus rank that social specialists are found younger than social generalists?

Response: The figure shows that within families, specialist genera are younger than generalists. Our previous figure said 'per family' which we agree is difficult to interpret, so we changed it to 'within families' to get this across.

16- "because genome size estimates are often based on cultivated microbes that differ markedly from environmentally derived genomes, previous suggestions of a positive correlation between genome size and niche range are likely biased": please rephrase this sentence, as every work has its own inherent bias, including the present work.

Response: Yes, this wording was arguably a bit arrogant. We have rephrased to: "previous suggestions of a positive correlation between genome size and niche range may have overlooked the contrasting correlations."

17- "...for example, the genus *Polyangium* with a mean genome size of 12.7Mb." Can you add the SNB score too, so it reads "for example the genus *Polyangium* with a mean genome size of 12.7Mb and a SNB of XX".

Response: We rewrote to: "Polyangium with a mean genome size of 12.7Mb and an SNB of 0.21 (Z-score = -1.36)."

Methods

18- "Taxa that were represented by less than 5 reads were removed before dissimilarity calculations": 5 reads in the total dataset or per sample?

Response: Per sample, we added it.

19- "For some of the ecological dissimilarity": which ones?

Response: We rewrote to: "For the ecological dissimilarity calculations that take number of reads into account". We originally had 'some' in there because for not all these measures it matters whether making the number of reads relative matters, for example for the correlation coefficients it does not matter if this is based on absolute abundances or relative abundances. But for the exemplified Bray-Curtis dissimilarity and Aitchison distance it does.

20- "pairwise comparisons can be based on few taxa": Do you mean that it is the case "pairwise comparisons was based on a reduced number taxa", or that even with a reduced number of taxa the analysis is robust or meaningful "pairwise comparisons can be computed albeit based on few taxa"?

Response: We mean the first, and have rewritten to: "Because the taxonomic profiles contain many unknowns at lower ranks, pairwise comparisons are sometimes based on only few taxa."

Displays

21- "Both colour-coding and size of markers represents the Shannon entropy": Why depicting the same metrics twice?

Response: The only reason is because it's easier to see.

22- "Blue lines indicate samples with low alpha diversity and yellow lines indicate samples with high alpha diversity": Would it be possible to add a color scale as the alpha-diversity is here represented by a color gradient.

Response: Yes, we have added the colour gradient, also in Fig. 4d which was also missing.

23- "Annotated biomes are arranged according to alpha diversity": is it the cumulative alpha-diversity of all samples of the biome (Boolean 'or') or the average alpha diversity?

Response: It's the mean alpha diversity and we have clarified wherever needed throughout the text.

Reviewer #3 (Remarks to the Author):

The authors present a very interesting and original study that uses the distribution of taxa with respect to overall microbiome composition in diverse natural habitats on a large scale as a proxy for their ecological niche breadth. They take advantage of the huge amount of existing data to bring some standard approaches (e.g. the study of co-occurrence of different species/taxa) to an entirely new scale, both data-wise and conceptually. This results, I believe, in an important empirical

contribution. It uncovers many new patterns, some of them expected (larger pan-genome in generalists than in specialists) and other, largely unexpected and surprising given common assumptions in the field, such as the dominance of generalists in terms of abundance. Overall, the methodology is state of the art, and its robustness is shown using appropriate controls, random permutation tests, and comparison to alternative technical solutions.

The study is largely descriptive, and I tend to favor more hypothesis-driven analyses. But I do believe that in this case, the novelty of the observed patterns has an important value by itself, and studying the principles and mechanisms that produce them would require substantial work that is probably beyond the scope of this study. For instance, the authors discuss the differences in genomic content and size between specialists and generalists in low vs high diversity communities, as well as functional enrichment patterns, to a somewhat coarse-grained extent, which is probably a limitation imposed by the large scale of the study.

Response: We agree with the reviewer and do plan follow-up studies.

In summary, I enjoyed reading the paper. I think it is an important contribution to the field and opens many new questions and opportunities for future work. If any criticism, I found some figures very data-dense, which somewhat hinders their straightforward visual interpretation (e.g. fig 4d or Fig 5). Especially in a study of this scale, I like figures that summarize the main, big-picture findings in a more graphically interpretable way (with more detailed figures and of course the data itself available as a supplement to those that want to delve into the details). But this is a matter of personal choice, and of course, it is up to the authors how they present their data.

Response: We have increased the size of most panels by 25% to at least make the 'multitude of dots' a little easier to interpret. In addition, we have changed Fig. 5 to a format that we believe is better interpretable based on a suggestion from reviewer 2.

Djordje Bajic

Reviewer #4 (Remarks to the Author):

The manuscript by von Meijenfeldt and colleagues proposes a community diversity-based metric for identifying ecological generalists and specialists. The main idea is that species living in very similar communities are likely to be specialists while those living in communities with diverse compositions are generalists. The underlying assumption is that the community diversity is an indicator of the environmental selection pressures. Once this definition is accepted, the authors use it to investigate distinguishing features of generalists compared to those of the specialists. This revealed differences in abundance profiles (generalists "stochastically" dominating while specialists being at low but constant abundance) as well as in their pangenome structures.

Overall, we find the topic highly interesting and of potential broad interest. The conclusions, however, require much stronger support.

Main points:

1. The central pillar of the study is the metric termed 'social niche breadth' (SNB). Yet, the manuscript

is quite shallow on the justification of the used formula and its potential pitfalls. A rank correlation based metric needs to be used with much caution when several members of the community are very low abundance (a typical case in many habitats including soil). In such cases, the technical artifacts (DNA extraction, amplification, data processing etc.) mean that the estimated abundances are only order of magnitude estimates with ranks having no real meaning. How sensitive are the results to use if this metric? Using a combination of different dissimilarity metrics will be much more robust.

Response: We agree with the reviewer that a rank-based correlation measure has downsides. For this reason we benchmarked 150 different pairwise dissimilarity measures, many of which are not correlation-based. Perhaps surprisingly, it turned out that the inverse Spearman's rank correlation was the best to separate the annotated biomes (Supp. Fig. 6a), although other metrics give quite similar results. We believe that at the global scale the rank of taxa shows the least noise because the distance is maximal when no taxa overlap, while if there are some overlapping taxa their relative rank becomes important. Importantly, other measures like example Bray-Curtis, Jaccard, or UniFrac showed relatively good correlations with our chosen method (Supp. Fig. 6b). We have rewritten the Supplementary Results and Discussion to get this point across and have added this sentence to the Results section of the main text: "After benchmarking 150 different ecological dissimilarity measures for their ability to separate the annotated biomes, we chose mean pairwise dissimilarity based on the inverse Spearman's rank correlation of known taxa at taxonomic rank order to quantify SNB (Supp. results and discussion)."

2. Another potential confounder is the niche alteration by microbes – many microbes secrete small molecules like amino acids that can alter the niches. The implications of such metabolic interactions on the analysis should be discussed as it can underestimate the true habitat diversity.

Response: We agree and we have now made it clear in the introduction that we assume to look at the realised niche that includes both environmental and microbial interactions like cross-feeding:

"Rather than the theoretical fundamental niche, microbial occurrence represents its empirical realised niche. Because of complex interactions within microbial communities, the realised niche can both be smaller, e.g. due to competition {10.15698/mic2018.05.628}, or larger, e.g. due to metabolic dependencies {10.1038/s41559-021-01505-0}."

And

"As such, we treat community composition as a proxy for the realised niche of a microbe, that reflects both the abiotic environment and the microbial interactions within. Similar reasoning has been used to quantify niche range of Eukaryotes without the use of external habitat definitions {10.1111/j.1365-2745.2007.01236.x}."

3. The manuscript seems to question the "classic" ecological theory that specialists have fitness advantage at the expense of versatility. Yet, this should be more prominently discussed. What explanations the data/analysis offers against the classical view? Both sides of the arguments should be clearly laid out and potential pitfalls discussed. An orthogonal analysis of longitudinal datasets will be important in this context.

Response: We agree that this is one of the most interesting points of our paper. Although a deeper investigation of this observation is difficult because the (longitudinal) data is missing, we have added to better highlight this point:

“Our data counter the classic “jack of all trades” argument, which suggested that specialist should have a local fitness advantage at the expense of ecological versatility. We explored possible explanations for the local dominance of generalists over specialists and their relatively variable abundance. First, although generalist genera contained more species than specialists genera in the total dataset (Supp. Fig. 18a), we do not find evidence that they also contain a higher number of species within samples (Supp. Fig. 18a), but not that only a small fraction of genera could be taxonomically classified on the species rank (Supp. Fig. 18b). Alternatively, social niche breadth may reflect the classical distinction between r-strategists and K-strategists (vys).”

4. The choice of including just a few meta-transcriptomics datasets is puzzling, it does not add anything to the study but rather adds potential confounder. Further, it is unclear how 16S vs shotgun metagenomics estimates are pooled together. I recommend analysing the two separately to check the robustness of the conclusions.

Response: Thank you for the suggestion, we have added a new analysis ('Robustness to experiment type' in the Supplementary Results and Discussion) where we do this analysis. See also our earlier responses to other reviewers regarding the experiment types.

5. The manuscript is poor on clearly reporting the effect sizes, which are modest in many cases and thus potentially sensitive to the choices of the metrics and any biases in the data.

Response: While we agree that effect sizes are often small, we do not believe that our overall results are very sensitive to biases in the data. In the section 'Social niche breadth robustly reflects community heterogeneity' of the Supplementary Results and Discussion we extensively show that different kinds of noise do not substantially alter our conclusions. We believe that the results we find show up despite of many sources of noise, instead of because of the noise. We have added Supplementary Table 6 and Supplementary Table 7 to report the effect sizes.

6. Repetition of the main analyses in a subset of habitats would be very insightful in testing the robustness/generalisation of the conclusions drawn.

Response: We have now added the new analysis where we show robustness of our results in subsets of the data according to experiment type. In addition, the Supplementary Results and Discussion includes a comparison with only the human and marine subsets of the data (Supp. Fig. 12).

7. Given the pan-genome level differences between generalists and specialists, it becomes imperative to discuss whether the “true functional taxonomic unit” should be the sub-species/strain rather than species or higher.

Response: Yes, but our data does not allow us to do this, because most MGnify annotations do not go to species rank, so we only have a genus annotation. We have made this clearer in the text for example with

“Since the taxonomic annotations are based on a reference taxonomy and the biosphere is relatively unexplored, sometimes high-ranking taxa do not have low-ranking annotations like species (Fig. 1e).”

and

“but note that only a small fraction of genera could be taxonomically classified on the species rank”

and

"Although these genomes likely belong to different strains or species than those observed in MGnify, we reduce the inconsistencies in our analysis by assessing their genomic features at the genus rank."

8. The enrichment analyses in the last figure are way too descriptive without any insights. I suggest moving the long lists to supplementary while presenting a noteworthy summary in the main figure/text.

Response: We have changed Fig. 5 to something that we believe is better readable based on the suggestion of Reviewer 2.

9. Introduction would benefit from explicit mentioning of classic ecological theory re. niche occupation & competitive exclusion (terms of these theories are referred to in the Results section, such as r-strategists and K-strategists, but would be good to briefly address in introduction). A more substantial explanation of what falls under the term 'niche breadth' and the literature usage of this will also be important to include.

Response: We have added a section on the n-dimensional niche concept in the introduction to clarify how we envision the SNB score (as a proxy for the n-dimensional realised niche of a microbe that includes both environmental and intercommunity interactions):

"Niche breadth definitions that assess the full n-dimensional niche space [htj] have been based on occurrence in microbiomes. Rather than the theoretical fundamental niche, microbial occurrence represents its empirical realised niche. Because of complex interactions within microbial communities, the realised niche can both be smaller, e.g. due to competition {10.15698/mic2018.05.628}, or larger, e.g. due to metabolic dependencies {10.1038/s41559-021-01505-0}."

And

"As such, we treat community composition as a proxy for the realised niche of a microbe, that reflects both the abiotic environment and the microbial interactions within. Similar reasoning has been used to quantify niche range of Eukaryotes without the use of external habitat definitions {10.1111/j.1365-2745.2007.01236.x}."

In addition, we have added this section to the Results section:

"Different from studies that investigate co-occurrence of taxa across samples {10.1371/journal.pcbi.1002606,10.1038/s41559-020-01353-4}, SNB quantifies the range of communities that a taxon can occur in. SNB treats each sample as a local niche, and infers that taxa that occur across highly differing communities are social generalist, while taxa that occur in similar communities are social specialists. Since SNB is calculated when a taxon is present over a detection limit of 1/10,000 reads, relative abundance and associated variability of a taxon's distribution are observables and can be associated to niche range, rather than part of the definition as in {10.1890/08-0851.1}."

And mention competitive exclusion further on:

"Although the genomes of social generalists are large compared to those of LD- specialists (Fig. 4d), the genomes in samples with low alpha diversity are still moderate in size compared to the large genomes in samples with high alpha diversity (Fig. 4e). High alpha diversity samples contain many

specialists (Fig. 2e), and their high richness may be a driver of specialisation through competitive exclusion, as has been suggested to explain a negative correlation between niche breadth and local diversity in Eukaryotes (10.1111/geb.13011)."

10. "To test this hypothesis, we compared SNB of microbes to their predicted maximal growth rates": the growth rate prediction is a tricky business. Why not use growth rates calculated from metagenomic samples with tools like coptr, iRep etc.?"

Response: We used the EGGO database because growth rate estimates based on codon usage show better results than those based on peak to trough ratios, see (<https://doi.org/10.1038/s41396-020-00773-1>).

11. Terms like "low diversity specialists" are unclear and warrant clear definitions.

Response: We agree and have rewritten the results section to make it clearer, including definitions.

12. Figure captions are sometimes unclear (e.g. Fig.2 panel e: what is the key for the letters?). Sometimes citations to the supplementary are spelled out completely or shortened. Consistency might be good.

Response: We have added the keys for the letters or written out full (as they are the first letters of taxonomic ranks). In addition, we have added colour bars to figures where they were missing. We have changed all 'Supplementary' to 'Supp.'.

13. Regarding the genomes in generalist genera being more variable in size, what is the hypothesis? What is the literature suggesting? What functions are coded on these variable regions?

Response: We added: "We hypothesize that the observed genomic flexibility allows members of generalist genera to rapidly acquire the genes needed to thrive in a given local environment (10.1128/mBio.00335-10,10.1038/nrmicro2795), their higher growth rate potential allowing them to outgrow specialists."

14. Please consider making a github repository to share the code and the results

All code used for the manuscript is available upon request. In addition, we will put a stand-alone script to calculate SNB on <https://github.com/MGXlab> upon acceptance.

15. "Lachnospiraceae (SNB = 0.50) and its genera (Supp. table 3), obligate anaerobes that were previously regarded as specialists 23 also have a high SNB" :Why Lachnospiraceae have a high SNB, where the diversity of their habitats come from? In general these concepts of "high diversity habitats" and "low diversity habitats" are not completely clear.

Response: We have now clarified what we mean with high and low throughout the manuscript by using a modified Z score for each taxonomic rank (see also response to Reviewer 1). The large variation of habitats in which Lachnospiraceae is found is something we observe, as it is reflected in its high SNB. We do want to stress that is found in a lot of samples, and we have updated the text accordingly:

"The family Lachnospiraceae (Z-score=1.74, which is found in over half of all samples, n = 11,887) and its genera (median Z-score = 0.79), obligate anaerobes that were previously regarded as habitat specialists {10.1038/s41467-017-01265-1} also have a high SNB for their ranks, highlighting the heterogeneity of the communities where they are found."

16. "(but see 48)" Please state explicitly what authors what to point out fro the reference.

Response: We have rewritten to: "however the genus may be ubiquitous in part because it is a common contaminant of DNA extraction kits {10.1186/s12915-014-0087-z}".Decision Letter, first revision:

22nd December 2022

Dear Dr. Dutilh,

Thank you for submitting your revised manuscript "On specialists and generalists: niche range strategies throughout the tree of life" (NATECOLEVOL-220717078A). It has now been seen again by the original reviewers 2-4 and their comments are below. The reviewers find that the paper has improved in revision, and therefore we'll be happy in principle to publish it in Nature Ecology & Evolution, pending minor revisions to satisfy the reviewers' final requests and to comply with our editorial and formatting guidelines.

[REDACTED]

Reviewer #2 (Remarks to the Author):

The article, now entitled "On specialists and generalists: niche range strategies across throughout the tree of life" has been diligently revised. New analyses, displays and methods are reported. At this stage, I only have a few additional minor comments:

Minor comments

- 1- Abstract line 13-14 "specialists are restricted to a single environment". If it is about the 'traditional' definition of specialist, I would rather argue that they are restricted to narrow *conditions*, albeit these conditions can be found in diverse environments as long as micro-niches in these biomes display these restricted conditions. Otherwise, it should be "habitat specialists are restricted to a single environment".
- 2- Line 36-37 "Niche breadth definitions that assess the full n-dimensional niche space have been based on occurrence in microbiomes". This sentence remains unclear to me.
- 3- Line 141: "small phyla": does it mean low diversity (LD) phyla?
- 4- Line 157: "Taxa with high and low SNB are dispersed throughout the prokaryotic tree of life (Fig.

352d),”: fig 2d is representing z-score and not directly SNB.

5- Line 194 and 262 *social* generalist?

6- Line.268-269: “It was previously suggested that generalist species are evolutionary younger than specialist species 27”. I think here, it is habitat specialist and habitat generalist

7- Figure 1 (i) panel c: can the sample be ordered to have a Gaussian shape? Or why is the curve irregular? (ii) Panel f “Annotated biomes with high mean alpha diversity have low beta diversity.”

Albeit true for the biome with extreme high alpha diversity and extreme high beta diversity, there is also biomes with both low alpha and beta diversity.

Supplementary

1 Line 39: “social niche 39 breadt” should read breadth

Reviewer #3 (Remarks to the Author):

On my side, everything adressed, good to go.

Reviewer #4 (Remarks to the Author):

The response is generally satisfactory. My only remaining concern is that "available upon request" for data/code is not a good scientific practice.

Our ref: NATECOLEVOL-220717078A

18th January 2023

Dear Dr. Dutilh,

Thank you for your patience as we’ve prepared the guidelines for final submission of your Nature Ecology & Evolution manuscript, "On specialists and generalists: niche range strategies throughout the tree of life" (NATECOLEVOL-220717078A). Please carefully follow the step-by-step instructions provided in the attached file, and add a response in each row of the table to indicate the changes that

36you have made. Please also check and comment on any additional marked-up edits we have proposed within the text. Ensuring that each point is addressed will help to ensure that your revised manuscript can be swiftly handed over to our production team.

****We would like to start working on your revised paper, with all of the requested files and forms, as soon as possible (preferably within two weeks). Please get in contact with us immediately if you anticipate it taking more than two weeks to submit these revised files.****

In recognition of the time and expertise our reviewers provide to Nature Ecology & Evolution's editorial process, we would like to formally acknowledge their contribution to the external peer review of your manuscript entitled "On specialists and generalists: niche range strategies throughout the tree of life". For those reviewers who give their assent, we will be publishing their names alongside the published article.

Nature Ecology & Evolution offers a Transparent Peer Review option for new original research manuscripts submitted after December 1st, 2019. As part of this initiative, we encourage our authors to support increased transparency into the peer review process by agreeing to have the reviewer comments, author rebuttal letters, and editorial decision letters published as a Supplementary item. When you submit your final files please clearly state in your cover letter whether or not you would like to participate in this initiative. Please note that failure to state your preference will result in delays in accepting your manuscript for publication.

Cover suggestions

As you prepare your final files we encourage you to consider whether you have any images or illustrations that may be appropriate for use on the cover of Nature Ecology & Evolution.

Nature Ecology & Evolution has now transitioned to a unified Rights Collection system which will allow our Author Services team to quickly and easily collect the rights and permissions required to publish your work. Approximately 10 days after your paper is formally accepted, you will receive an email in providing you with a link to complete the grant of rights. If your paper is eligible for Open Access, our Author Services team will also be in touch regarding any additional information that may be required to arrange payment for your article.

Please note that *Nature Ecology & Evolution* is a Transformative Journal (TJ). Authors may publish their research with us through the traditional subscription access route or make their paper immediately open access through payment of an article-processing charge (APC). Authors will not be required to make a final decision about access to their article until it has been accepted. [Find out more about Transformative Journals](https://www.springernature.com/gp/open-research/transformative-journals)

Authors may need to take specific actions to achieve [compliance with funder and institutional open access mandates](https://www.springernature.com/gp/open-research/funding/policy-compliance-faqs). If your research is supported by a funder that requires immediate open access (e.g. according to [Plan S principles](https://www.springernature.com/gp/open-research/plan-s-compliance)) then you should select the gold OA route, and we will direct you to the compliant route where possible. For authors selecting the subscription publication route, the journal's standard licensing terms will need to be accepted, including [self-archiving-and-license-to-publish](https://www.nature.com/nature-portfolio/editorial-policies/self-archiving-and-license-to-publish). Those licensing terms will supersede any other terms that the author or any third party may assert apply to any version of the manuscript.

[REDACTED]

[REDACTED]

Reviewer #1:

Remarks to the Author:

I appreciate author's thorough revisions of the manuscript and supplementary information to improve reproducibility.

After reading the authors response to my previous comment regarding the "candidatus" taxa I specifically find this justification not too accurate. Many candidatus taxa are mainly lacking the culture to get a valid description and could be among highly abundant and ubiquitous lineages as well. I would recommend authors to remove this section (Line 145-156) as it is not among the main findings of their study and relies highly on the correct designation of candidatus status that might not be reliably annotated and updated in different databases.

Reading the revised version of the manuscript, I suggest some minor edits and changes:

Line 26- social niche

Line 60- quantify social niche range

I like the expanded explanation in lines 49 -61.

Line 141-144- this sentence reads a bit confusing. Please rephrase to explain the impact of taxon rank jump on SNB clearer.

Line 370- the evolutionary history of different types based on SNB is not specifically the focus of this study and I would recommend authors to rephrase this sentence "Generalist genera are older than specialist genera"

Supplemental data

Line 39- fix the typo

Reviewer #2:

Remarks to the Author:

The article, now entitled "On specialists and generalists: niche range strategies across throughout the tree of life" has been diligently revised. New analyses, displays and methods are reported. At this stage, I only have a few additional minor comments:

Minor comments

1- Abstract line 13-14 "specialists are restricted to a single environment". If it is about the 'traditional' definition of specialist, I would rather argue that they are restricted to narrow *conditions*, albeit these conditions can be found in diverse environments as long as micro-niches in these biomes display these restricted conditions. Otherwise, it should be "habitat specialists are restricted to a single environment".

39- 2- Line 36-37 "Niche breadth definitions that assess the full n-dimensional niche space have been based on occurrence in microbiomes". This sentence remains unclear to me.
- 3- Line 141: "small phyla": does it mean low diversity (LD) phyla?
- 4- Line 157: "Taxa with high and low SNB are dispersed throughout the prokaryotic tree of life (Fig. 2d),": fig 2d is representing z-score and not directly SNB.
- 5- Line 194 and 262 *social* generalist?
- 6- Line.268-269: "It was previously suggested that generalist species are evolutionary younger than specialist species 27". I think here, it is habitat specialist and habitat generalist
- 7- Figure 1 (i) panel c: can the sample be ordered to have a Gaussian shape? Or why is the curve irregular? (ii) Panel f "Annotated biomes with high mean alpha diversity have low beta diversity." Albeit true for the biome with extreme high alpha diversity and extreme high beta diversity, there is also biomes with both low alpha and beta diversity.

Supplementary

1 Line 39: "social niche 39 breadt" should read breadth

Reviewer #3:

Remarks to the Author:

On my side, everything adressed, good to go.

Reviewer #4:

Remarks to the Author:

The response is generally satisfactory. My only remaining concern is that "available upon request" for data/code is not a good scientific practice.

Author Rebuttal, first revision:

Reviewer #1:

Remarks to the Author:

I appreciate author's thorough revisions of the manuscript and supplementary information to improve reproducibility.

Thank you!

After reading the authors response to my previous comment regarding the "candidatus" taxa I specifically find this justification not too accurate. Many candidatus taxa are mainly lacking the culture to get a valid description and could be among highly abundant and ubiquitous lineages as well. I would recommend authors to remove this section (Line 145-156) as it is not among the main findings of their

40study and relies highly on the correct designation of candidatus status that might not be reliably annotated and updated in different databases.

We agree that the definition of ‘candidatus’ taxa is that they are not yet validly described, e.g. because they lack a culture isolate. While in theory they could be among highly abundant and ubiquitous lineages, we do observe that they have a relatively low SNB and would like to mention this. We would like to keep the discussion of ‘candidatus’ phyla in, but we have toned it down into:

“Many phyla with the “candidatus” status have a low SNB compared to validly described phyla (Supp. Fig. 17). The connection between the candidatus status and low SNB may reflect a discovery bias of these phyla where widespread lineages (generalists) tend to be discovered and described sooner than rare ones (specialists). Specialised organisms in candidate phyla may require specific growth conditions, as reflected in their relatively consistent social communities and low SNB. In addition, several candidate phyla including the bacterial Candidate Phyla Radiation and DPANN archaea may consist of obligate symbionts of specific hosts {10.1016/j.cell.2018.02.016}. Whereas it was recently shown that consortia of obligate symbionts can grow on a wider range of carbon sources than their individual members and thus expand their metabolic niche {10.1038/s41559-021-01505-0}, the individual microbes in these consortia are social specialists as they require specific partners in their local communities.”

Reading the revised version of the manuscript, I suggest some minor edits and changes:

Line 26- social niche

We have rewritten this sentence to “Together, our analysis shines data-driven light on microbial niche range strategies.” We agree that we analyse ‘social’ niche range but we believe that for the final sentence of the abstract it’s important to mention that this new approach shines new light on niche range strategies ‘in general’ and we thus did not add ‘social’ here.

Line 60- quantify social niche range

We have added ‘social’.

I like the expanded explanation in lines 49 -61.

Thank you! 😊

Line141-144- this sentence reads a bit confusing. Please rephrase to explain the impact of taxon rank jump on SNB clearer.

We agree and have rewritten to: “There are many phyla that have low SNB scores and contain few classes, orders, and families compared to the dominant ones described above. These low-scoring phyla contain relatively few described classes, orders, and families so we observe that the distribution of SNB scores is more skewed towards social specialism at the phylum rank (median SNB = 0.36) than at those lower ranks (median SNB = 0.38-0.43, see Fig. 2d).”

Line 370- the evolutionary history of different types based on SNB is not specifically the focus of this study and I would recommend authors to rephrase this sentence “Generalist genera are older than specialist genera”

We have rewritten to remove the focus on the evolutionary process but rather on the observation:

“Generalist genera are older than specialist genera, and have large and open pan genomes with which they have adapted to different habitats.”

Supplemental data

Line 39- fix the typo

Thanks, we have fixed the typo.

Reviewer #2:

Remarks to the Author:

The article, now entitled “On specialists and generalists: niche range strategies across throughout the tree of life” has been diligently revised. New analyses, displays and methods are reported. At this stage, I only have a few additional minor comments:

Minor comments

1- Abstract line 13-14 “specialists are restricted to a single environment”. If it is about the ‘traditional’

42definition of specialist, I would rather argue that they are restricted to narrow *conditions*, albeit these conditions can be found in diverse environments as long as micro-niches in these biomes display these restricted conditions. Otherwise, it should be “habitat specialists are restricted to a single environment”.

We agree and have rewritten the next sentence to clarify:

“Although a classical concept in ecology, niche breadth has remained challenging to quantify for microbes because it depends on an objective definition of the environmental conditions.”

2- Line 36-37 “Niche breadth definitions that assess the full n-dimensional niche space have been based on occurrence in microbiomes”. This sentence remains unclear to me.

We have rewritten to:

“The full n-dimensional niche space of organisms {htj} has previously been assessed based on their occurrence in different environments.”

3- Line 141: “small phyla”: does it mean low diversity (LD) phyla?

We have rewritten this section to:

“There are many phyla that have low SNB scores and contain few classes, orders, and families compared to the dominant ones described above. These low-scoring phyla contain relatively few described classes, orders, and families so we observe that the distribution of SNB scores is more skewed towards social specialism at the phylum rank (median SNB = 0.36) than at those lower ranks (median SNB = 0.38-0.43, see Fig. 2d).”

4- Line 157: “Taxa with high and low SNB are dispersed throughout the prokaryotic tree of life (Fig. 2d),”: fig 2d is representing z-score and not directly SNB.

Yes we agree, and have rewritten to:

“Taxa with high and low Z-scores are dispersed throughout the prokaryotic tree of life (Fig. 2d).”

5- Line 194 and 262 *social* generalist?

For line 194, yes we have rewritten. Line 262 did not contain 'generalists'.

6- Line.268-269: "It was previously suggested that generalist species are evolutionary younger than specialist species 27". I think here, it is habitat specialist and habitat generalist

Yes you are right, we have rewritten to:

It was previously suggested that habitat generalist species are evolutionary younger than habitat specialist species {10.1038/s41467-017-01265-1}.

7- Figure 1 (i) panel c: can the sample be ordered to have a Gaussian shape? Or why is the curve irregular? (ii) Panel f "Annotated biomes with high mean alpha diversity have low beta diversity." Albeit true for the biome with extreme high alpha diversity and extreme high beta diversity, there is also biomes with both low alpha and beta diversity.

(i) The distribution is cut off at 50,000 reads, because this is one of our selection criteria, which is why the shape is asymmetrical. We have added this cut-off in the figure itself, and have changed the caption to make this more clear:

"Total number of taxonomically annotated reads per samples (n = 22,518 samples). Boxplot shows interquartile range and median. No samples with less than 50,000 reads were selected."

(ii) We very much agree with your observation about the alpha and beta diversity and have added in the caption of the figure:

"Annotated biomes with high mean alpha diversity have low beta diversity, whereas both low and high beta diversity is found among annotated biomes with low mean alpha diversity."

Supplementary

441 Line 39: "social niche 39 breadt" should read breadth

Thanks, we have fixed the typo.

Reviewer #3:

Remarks to the Author:

On my side, everything adressed, good to go.

Thanks for this! 😊

Reviewer #4:

Remarks to the Author:

The response is generally satisfactory. My only remaining concern is that "available upon request" for data/code is not a good scientific practice.

We agree and will add all the code (including the code for making the figures) to a Zenodo DOI.

Final Decision Letter:

27th February 2023

Dear Dr Dutilh,

We are pleased to inform you that your Article entitled "A social niche breadth score reveals niche range strategies of generalists and specialists", has now been accepted for publication in Nature Ecology & Evolution.

Over the next few weeks, your paper will be copyedited to ensure that it conforms to Nature Ecology and Evolution style. Once your paper is typeset, you will receive an email with a link to choose the appropriate publishing options for your paper and our Author Services team will be in touch regarding any additional information that may be required

After the grant of rights is completed, you will receive a link to your electronic proof via email with a request to make any corrections within 48 hours. If, when you receive your proof, you cannot meet

45this deadline, please inform us at rjsproduction@springernature.com immediately.

You will not receive your proofs until the publishing agreement has been received through our system

Due to the importance of these deadlines, we ask you please us know now whether you will be difficult to contact over the next month. If this is the case, we ask you provide us with the contact information (email, phone and fax) of someone who will be able to check the proofs on your behalf, and who will be available to address any last-minute problems . Once your paper has been scheduled for online publication, the Nature press office will be in touch to confirm the details.

Acceptance of your manuscript is conditional on all authors' agreement with our publication policies (see www.nature.com/authors/policies/index.html). In particular your manuscript must not be published elsewhere and there must be no announcement of the work to any media outlet until the publication date (the day on which it is uploaded onto our web site).

Please note that *Nature Ecology & Evolution* is a Transformative Journal (TJ). Authors may publish their research with us through the traditional subscription access route or make their paper immediately open access through payment of an article-processing charge (APC). Authors will not be required to make a final decision about access to their article until it has been accepted. [Find out more about Transformative Journals](https://www.springernature.com/gp/open-research/transformative-journals)

Authors may need to take specific actions to achieve [compliance](https://www.springernature.com/gp/open-research/funding/policy-compliance-faqs) with funder and institutional open access mandates. If your research is supported by a funder that requires immediate open access (e.g. according to [Plan S principles](https://www.springernature.com/gp/open-research/plan-s-compliance)) then you should select the gold OA route, and we will direct you to the compliant route where possible. For authors selecting the subscription publication route, the journal's standard licensing terms will need to be accepted, including [self-archiving-and-license-to-publish](https://www.nature.com/nature-portfolio/editorial-policies/self-archiving-and-license-to-publish). Those licensing terms will supersede any other terms that the author or any third party may assert apply to any version of the manuscript.

An online order form for reprints of your paper is available at <https://www.nature.com/reprints/author-reprints.html>. All co-authors, authors'

46institutions and authors' funding agencies can order reprints using the form appropriate to their geographical region.

We welcome the submission of potential cover material (including a short caption of around 40 words) related to your manuscript; suggestions should be sent to Nature Ecology & Evolution as electronic files (the image should be 300 dpi at 210 x 297 mm in either TIFF or JPEG format). Please note that such pictures should be selected more for their aesthetic appeal than for their scientific content, and that colour images work better than black and white or grayscale images. Please do not try to design a cover with the Nature Ecology & Evolution logo etc., and please do not submit composites of images related to your work. I am sure you will understand that we cannot make any promise as to whether any of your suggestions might be selected for the cover of the journal.

You can generate the link yourself when you receive your article DOI by entering it here: <http://authors.springernature.com/share>.

[REDACTED]

P.S. Click on the following link if you would like to recommend Nature Ecology & Evolution to your librarian <http://www.nature.com/subscriptions/recommend.html#forms>

** Visit the Springer Nature Editorial and Publishing website at http://editorial-jobs.springernature.com?utm_source=ejp_NEcoE_email&utm_medium=ejp_NEcoE_email&utm_campaign=ejp_NEcoE for more information about our career opportunities. If you have any questions please click [here](mailto:editorial.publishing.jobs@springernature.com).**